# Structural, Optical and Dynamic Properties of Thin Smectic Films

**Izabela Śliwa [1] and A. V. Zakharov [2,*]**

[1]  Department of mathematical economics, Poznan University of Economics and Business, Al. Niepodleglosci 10, 61-875 Poznan, Poland; izabela.sliwa@ue.poznan.pl
[2]  Saint Petersburg Institute for Machine Sciences, The Russian Academy of Sciences, Saint Petersburg 199178, Russia
*  Correspondence: alexandre.zakharov@yahoo.ca

**Abstract:** The problem of predicting structural and dynamic behavior associated with thin smectic films, both deposited on a solid surface or stretched over an opening, when the temperature is slowly increased above the bulk transition temperature towards either the nematic or isotropic phases, remains an interesting one in the physics of condensed matter. A useful route in studies of structural and optical properties of thin smectic films is provided by a combination of statistical–mechanical theories, hydrodynamics of liquid crystal phases, and optical and calorimetric techniques. We believe that this review shows some useful routes not only for the further examining of the validity of a theoretical description of thin smectic films, both deposited on a solid surface or stretched over an opening, but also for analyzing their structural, optical, and dynamic properties.

**Keywords:** Liquid crystals; smectics; free-standing liquid crystal systems; smectic membranes

## 1. Introduction

Free-standing smectic films (FSSFs) and lipid membranes are curious and ubiquitous fluid-like objects in the realm of soft-matter science. In the long-scale limit they can be considered to be a new class of two-dimensional state embedded into three-dimensional space. Here the free-standing liquid crystal (LC) systems, composed of a stack of smectic layers confined by surrounding air [1] or water [2], or deposited on a solid surface, will be considered. Since there is no substrate, these freely supported fluid-like films represent an excellent model of low-dimensional systems for the study of surface effects as the film thickness is reduced.

It should be pointed out that in contrast to thin fluid-like films, there is a parallel world of crystalline [3], glassy freely supported membranes [4], and graphene films [5], but we will focused here on the fluid-like films composed of nano-sized sheets of LC matter.

Competition between surface and finite-sized effects leads to unusual physical properties of LC systems with a finite number of layers. One of the most interesting features of such LC systems is that under appropriate conditions they can be spread across an opening to form FSSFs which can be considered to be a stack of smectic layers confined by the surrounding air or water. Both meniscus, which connects the film with the solid frame, and the surface tension of the film are responsible for stability of these films. Moreover, the presence of the surface tension is believed to be responsible for intriguing surface ordering phenomena exhibited by these films. Unlike the preferential surface melting exhibited by conventional solids, the FSSF/air or FSSF/water interface appears to enhance the order of the surface layers so that they become ordered at temperatures well above the bulk smectic-A-isotropic (AI) [1] or smectic-A-nematic (AN) [6] transition temperatures $T_{\text{AI(N)}}$(bulk). This interesting phenomena has been reported both for LC compounds exhibiting

a first-order phase transition from smectic-A (SmA) to the isotropic (I) phase [1], as well as for compounds which possess a second-order smectic-A-nematic (N) phase transition [6]. In both types of systems, the central smectic layers of films begin to melt under increasing temperature above $T_{AI(N)}$(bulk). Thus, melting originates in the interior of the film and penetrates towards surfaces. Accordingly, at each step of the layer-thinning process, one can observe the coexistence of different LC phases, separated by phase limits (fronts) which move under temperature changes. The movement of these phase fronts consists of successive spontaneous layer-by-layer thinning transitions (local transformations of smectic layers into nematic and isotropic phases) of films, when the interior layer(s) is(are) squeezed-out in meniscus. Thinning was observed for different low-temperature smectic systems, such as SmA, SmC, and ferroelectric SmC*, but we will focus here on the review of the structural and dynamic properties of thin free-standing smectic-A films in air or water, as well as deposited on the solid surface.

A great variety of phenomena, both structural and dynamic, can be observed in thin FSSFs: layer-thinning transitions [1,2,6], stepwise reduction of heat capacity and reflectivity [1,6–8], sawtooth behavior of the surface tension upon heating the FSSFs above $T_{AI(N)}$(bulk) [9–12], dynamics of removal of one or several layers from the $N$-layer FSSF during the layer-thinning process [13–18], et al. In particular, it was shown that the AI or AN transitions occurs through a series of layer-thinning transitions, causing the films to thin in a stepwise manner as the temperature is increased above $T_{AI(N)}$(bulk) [1,2], whereas the film tension, at each thinning, abruptly drops to a lower value and then continues to increase with a smaller slope [9–12]. These transitions have been attributed to the reduction of smectic fluctuations in the bounding layers [7,19–21]. Associated phenomena have been observed in the capacities $C_p$ near the layer-thinning transition temperatures. These have been studied in different compounds [1,22,23] and consist of stepwise reductions of the value of $C_p(T)$ as the temperature $T$ is raised above $T_{AI(N)}$(bulk). Both by the calorimetric studies [1,2] and the theoretical investigations [7,23] it has been shown that the reduction of $C_p(T)$ is associated with the reduction of the number of layers in the film.

In turn, according to experimental results obtained by applying the technique of fluorescence scanning laser confocal microscopy [24], orientational and translational order of LCs across confining films can be much more complex than that predicted theoretically by mean-field modeling [7,25,26]. This may suggest that complex molecular orderings in thin LC films, both deposited on the solid surfaces or stretched over an opening, can appear as the result of the interplay between short-range forces and long-range intermolecular interactions. What concerns bounding and surface interactions, their significance for promoting layer alignment of molecules is obvious, although a detailed explanation of underlaying anchoring processes is still far from completeness. Furthermore, it is unclear whether these interactions should always be treated as if they were local. Indeed, whereas the surface interactions are usually considered to be being strictly local, experimental results indicate that they are nonlocal, albeit rather short-ranged [27]. On the contrary, the significance of long-range couplings between molecules for the appearance of ordered structures in LC systems has often been suggested, especially for the formation of the nematic phase [28]. Nevertheless, this question is still not clarified in detail. Then, the explanation of the role of nonlocal surface interactions and long-range intermolecular coupling in the organization of molecules in thin LC films is a challenging problem.

The understanding of how confinement influences the dynamics of the layer-thinning process when one or several layer(s) is(are) squeezed-out from the smectic film into the meniscus has relevance for several different areas of applied physics and material science. For description of the layer-thinning transition in FSSFs, several theoretical approaches have been suggested. First of all, it has been suggested that the squeezing-out transition is initiated by a thermally activated nucleation in which a density fluctuation forms a small circular hole (void) of critical radius in the smectic film [17,18], whereas in the second approach, the squeezing-out transition is initiated by spontaneous nucleation of dislocation loops, the growth of which causes a film to thin [15,16].

This review aims to show some useful routes not only for further examining the validity of theoretical descriptions of thin SmA films, both deposited on a solid surfaces or stretched over an opening, when the temperature is slowly increased above the bulk transition temperature towards either the nematic or isotropic phases, but also for analyzing their structural, optical, thermodynamic and dynamic properties.

## 2. Static Properties of Free-Standing Smectic Films and Molecular Ordering in Thin Liquid Crystal Films on a Solid Surfaces

There are several methods to investigate the coupling between structural and dynamic properties of thin smectic films, both suspended in air or deposited on the solid surface, and molecular structure.   A   usual route in these studies is provided by a combination of statistical–mechanical theories [7,8,11–13,16–21,23,25,26] and optical and calorimetric techniques [1,2,6,9,10,14,15,22,24,29–33].

### 2.1. Layer-Thinning Transitions and Optical Reflectivity Observed in FSSFs

One of the most effective techniques of the experimental investigation of thin LC films both deposited on solid substrates or in freely suspended films is the study of their optical properties, namely the optical transmission spectra [2,6] and the optical reflectivity [1,2,6], especially with the use of the techniques of fluorescence confocal microscopy [24,29] and X-ray reflectivity [30]. By measuring the optical reflectivity $\mathcal{R}$ of free-standing SmA films composed of certain LC compounds a remarkable phenomenon of layer-thinning transitions in FSSFs upon heating above $T_{AI(N)}$(bulk) has been revealed. During these transitions the FSSF with initial thickness of several tens of smectic layers can thin layer-by-layer and the film thickness reduction is suitably described by a power-law expression $N(t) \sim l^{-\nu}$, where $N$ is the number of layers, $l = \left( T_{AI(N)}(N) - T_{AI(N)}(bulk) \right) / T_{AI(N)}(bulk)$ is the reduced temperature, $T_{AI(N)}(N)$ is the temperature corresponding to the layer-thinning transition in $N$-layer smectic film, and $\nu$ is the critical exponent value belonging to the interval $[0.5, 0.9]$, for different compounds [31]. It has been shown both by calorimetric and optical studies that the free-standing LC films composed of 5-*n*-alkyl-2-(4-*n*-(*perfluoroalkyl-metheleneoxy*)*phenyl*) (H10F5MOPP) molecules may exhibit the following thinning sequence: $N = 15 \to 11 \to 9 \to 8 \to 7 \to 6 \to 5 \to 4 \to 3 \to 2$, as the film temperature is increased. The two-layer SmA film, composed of H10F5MOPP molecules ruptures in air at a temperature approximately 30 K above $T_{AI}$(bulk) [1] and the thinning transition is thermally driven and irreversible. For example, a two-layer film does not rupture for more than 5 h at 30 K above $T_{AI}$(bulk) and does not spontaneously thicken when cooled well into the SmA phase [32].

Only a small number of compounds possessing partially fluorinated alkyl chains show a layer-by-layer thinning process in air but films of *cyanobiphenyl's* compounds, for instance *nCB*, just rupture when heated above $T_{AI}$(bulk).  In turn, the thinning effect in thin smectic films, composed of decylcyanobiphenyls (10*CB*) and dodecylcyanobiphenyls (12*CB*) molecules in water on heating above $T_{AI}$(bulk) has been observed [2]. It was shown that these stable smectic films, composed of *cyanobiphenyl's* molecules, can be prepared in water with the help of a surfactant that induces homeotropic anchoring of LC molecules at the film/water interfaces. These films undergo a one-step thinning transition with the following thinning sequence: $N = 9 \to 8 \to 7 \to 6 \to 5 \to 4 \to 3 \to 2$, in which the film thickness decreases in a stepwise fashion until the film ruptures at $\sim$2 K above $T_{AI}$(bulk) [2]. This behavior is different from FSSFs in air: on heating to the isotropic phase the FSSFs in air either just rupture, for vast majority of LC compounds, or show a layer-by-layer thinning transition, for LC compounds possessing the partially fluorinated alkyl chains. Therefore, it clearly indicates that the film's environment plays a crucial role in formation of the thinning effect and FSSFs present unusual physical properties which are associated with the interplay of surface and finite-size effects.

Experimentally obtained values both for the reflectivity $\mathcal{R}(T)$ and heat capacity $C_p(T)$ decrease in a series of sharp steps separated by wide plateaus as the temperature is increased (see Figures 1 and 2, Ref. [1]). Moreover, the plateau values have been found to obey the laws $\mathcal{R}(T) \sim N^2$ ($N < 15$) and

$C_p(T) \sim N$. Therefore, it is clear that the steps correspond to discrete reductions of the film thickness, demonstrating the unique nature of this melting transitions. These transitions have been attributed to the reduction of smectic fluctuations in the bounding layers. It takes place because these films thin as the interior layers undergo the SmA-isotropic transition and the disjoining pressure [12] acting, for instance, across the $N$-layer and $(N-1)$-layer smectic film, is responsible for removal of one smectic layer from the $N$-layer smectic film into the surrounding reservoir during the layer-thinning process. It has also been shown, by means of high-resolution optical reflectivity investigations [32] that partially fluorinated compound, such as 2-4-(1,1-*dihydro*-2-(2-*perfluorobutoxy*) *perfluoroethoxy*) *phenyl*-5-*octyl pyrimidine* (H8F(4,2,1)MOPP), reveal a substantial compression of the smectic layer spacing during the layer-thinning transitions. It was found that upon heating the $N$-layer perfluorinated film ($N = 10, 9, 8, 7, ..., 3$) to its maximum temperature $T_{AI}(N)$ of existence, the average film layer thickness $\mathcal{L}$ decreases monotonically to a certain minimum value $\mathcal{L}_m$, and then, at the thinning transition to the $(N-1)$-layer film, $\mathcal{L}$ jumps to a nearly initial value. Upon further heating, the average smectic layer thickness in the new $(N-1)$-layer FSSF exhibits a similar behavior. It should be noted that a change in the average layer spacing can be as large as $\sim 0.1$ nm, and the minimum value of $\mathcal{L}$ in the $N$-layer film, which is reached at the temperature $T_{AI}(N)$, decreases with decreasing the number $N$ of the film layers. In other words, the minimum value of $\mathcal{L}$ in the nine-layer FSSF is smaller than in the ten-layer film, and $\mathcal{L}_m$ for eight-layer film is smaller than that in the nine-layer one, etc. These results are in contrast with data on the optical reflectivities of FSSF's made of a hydrogenated LC compounds, such as *n-pentyl*-4′-*pentanoyloxy-biphenyl*-4-*carboxylate* (54COOBC), composed of molecules with ordinary alkyl tails without fluorine atoms [33]. Though free-standing films of this material also undergo layer-thinning transitions upon heating above $T_{AI}(\text{bulk})$, their reflectivities, at given number $N$ of the film layers, do not change with increasing temperature up to its maximum value $T_{AI}(N)$. If the reflectivity of the $N$-layer FSSF does not change upon heating up to the temperature $T_{AI}(N)$ of its thinning transition, then the average layer thickness in this film is completely temperature independent. The origin of such diverse behavior of the smectic layers in FSSFs of different mesogens is not clear up to now.

The free-standing SmA films with two free bounding surfaces represent an excellent example of low-dimensional systems for the study of surface effects. Among other properties of such systems, research of surface tension behavior during the layer-thinning transitions is of both academic and applied interest. Recently, by means of the high-resolution calorimetric technique, direct measurements of the surface tension $\gamma$ of the partially fluorinated 5-*n-alkyl*-2-(4-*n-(perfluoroalkyl-metheleneoxy)phenyl*) films in air, during the layer-thinning transitions, has been carried out [10]. It has been found that at each thinning the film tension $\gamma$ abruptly drops to a lower value and then continues to increase with a smaller slope. This effect tends to repeat for the rest of the sequence of the layer-thinning transitions $N = 15 \rightarrow 11 \rightarrow 9 \rightarrow 8 \rightarrow 7 \rightarrow 6 \rightarrow 5 \rightarrow 4 \rightarrow 3 \rightarrow 2$ as the film temperature is increased, where each thinning is characterized by abrupt drops to lower values of $\gamma(T)$, after which it then continues to increase with a smaller positive slope.

Most of experimental techniques to study the effect of the molecular alignment of LC compounds in thin cells are focused on geometric aspects of the molecular anchoring and stabilization at solid surfaces. To achieve a better stabilization of LC mesophases and to gain a more precise control of their behaviors, various methods to prepare bounding surfaces have been developed [33–37]. The primary method was based on producing micro-grooves on polymer coated surface and, next, on using the rubbing technologies. By employing the atomic force microscopy, the rubbing process has significantly been improved to allow the formation of grooves on the microscale [34]. Alternative methods involve photolithography, enabling the imprinting of appropriate patterns onto photoactive polymer substrates [35], the technique of creating self-assembled alignment surfaces or the process of controlled self-assembly of LC molecules on surfaces [36], as well as the process of realigning of LC molecules at surfaces by means of the infrared laser beam [37].

Thus, experimental studies have shown that the introduction of fluorine functionality into a LC molecule often leads to the change of the observed behavior significantly from that of similar, fully hydroalkyl analogs. Exploring and understanding the mechanisms which connect the presence of fluorine with those molecules' novel macroscopic properties are therefore important both to advance technology and to clarify the relevant underlying physical interactions.

On the other hand, the theoretical treatment of both structural and dynamic properties of thin smectic films is not an easy task and requires a certain number of simplifying assumptions which may only be justified by comparison between model predictions and experiment. Thus, the combination of optical and calorimetric techniques with theoretical treatments provides a powerful tool for the investigation of both structural and dynamic properties of mesogenic compounds containing flexible moieties, especially partially fluorinated chains.

### 2.2. Formulation of the Mean-Field Model for the LC System Confined in Small Volume

### 2.2.1. The Surface Effect on the Molecular Ordering in thin LC Systems on a Solid Surface

In this Section it will be considered the LC system confined between two parallel surfaces. Both the structural and optical properties of such LC system will be considered in the framework of the mean-field model [25,26]. The smectic phase, consisting of $N$ smectic layers (each of the thickness $d$) oriented parallel to the bounding surfaces was considered to be the initial state of the LC system. The geometry of the LC system used for theoretical analysis is shown in Figure 1.

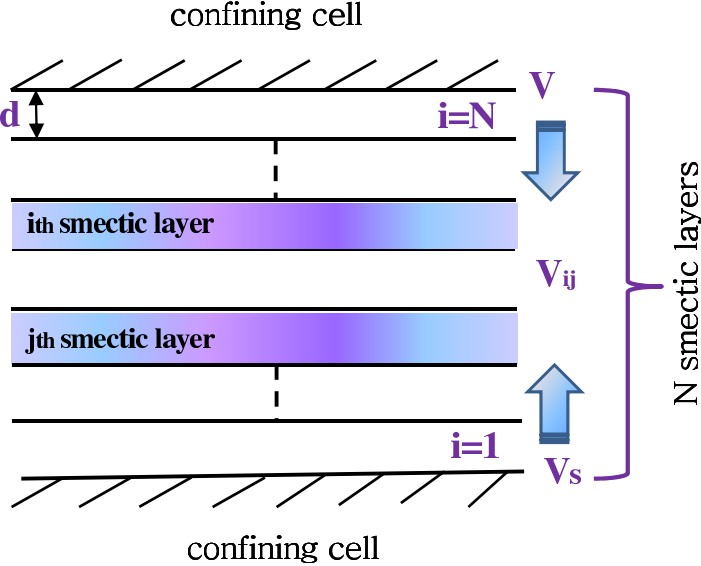

**Figure 1.** The geometry of the LC system used for theoretical analysis.

Two types of interactions will be considered, first, the short-range, rapidly decaying with distance, interactions of the smectic layers with confining surfaces ($V_i$ ($i = 0, s$)) and, second, the long-range van der Waals interactions between smectic layers ($V_{ij}$ ($i \neq j$)) [25,26]. Taking into account that the length of the smectic layers is much bigger than the thickness $Nd$, we can suppose that all the physical quantities depend only on the $z$-coordinate counted from the lower bounding surface.

The set of the effective anisotropic potentials $\Phi_i$ ($i = 1, ..., N$) within the $i$th smectic layer can be introduced in the framework of the mean-force approach [25,26]:

$$\Phi_i (z_i, \beta_i) = [V_i + \mathcal{A}_i + \alpha (V_i + \alpha \mathcal{B}_i) \cos (2\pi z_i)] P_2 (\cos \beta_i), \tag{1}$$

with

$$\mathcal{A}_i = \sum_{j \neq i}^{N} V_{ij} \eta_i, \quad \mathcal{B}_i = \sum_{j \neq i}^{N} V_{ij} \sigma_i, \tag{2}$$

where

$$\eta_i = \langle P_2 (\cos \beta_i) \rangle_i \tag{3}$$

is the local orientational order parameter (OP), while

$$\sigma_i = \frac{1}{\alpha} \langle P_2 (\cos \beta_i) \cos (2\pi z_i) \rangle_i \tag{4}$$

is the local translational order parameter, $P_2(\cos \beta_i)$ is the second-order Legendre polynomial, $\beta_i$ is the polar angle or the angle between the long axis of the molecule from the $i$-th layer and the director, whereas $\bar{z}_i = z/d$ is the dimensionless position of the molecule from the $i$-th layer, and the overbar has been (and will be) eliminated in the following equations. It should be noted that both these potentials $V_i$ and $V_{ij}$ describe the interactions of the $i$th layer with confining surfaces and the pair interactions between $i$th and $j$th molecules, respectively. Here $\langle (...) \rangle_i$ is the statistical–mechanical average with respect to the one-particle distribution function of the $i$th layer [7,38]

$$h_i (z_i, \beta_i) = \frac{1}{\mathcal{Q}_i} \exp \left[ \frac{\Phi_i (z_i, \beta_i)}{k_B T} \right], \tag{5}$$

where $T$ is the absolute temperature of the system, $k_B$ is the Boltzmann constant, $\mathcal{Q}_i = \int_0^1 dz_i \int_0^1 d \cos \beta_i \exp \left[ \frac{\Phi_i(z_i, \beta_i)}{k_B T} \right]$ is the partition function of the $i$th layer, respectively. Please note that in the smectic phase both OPs $\eta_i$ and $\sigma_i$ are nonzero, whereas in the nematic phase $\eta_i \neq 0$ and $\sigma_i = 0$, respectively. Finally, in the isotropic phase both OPs $\eta_i$ and $\sigma_i$ are equal to zero. Both sums $\mathcal{A}_i$ and $\mathcal{B}_i$ can be considered to be some weighted local order parameters. The constant $\alpha = 2 \exp[-(\pi r_0/d)^2]$ implicitly characterizes molecular packing within smectic layers, and $r_0$ is a characteristic length associated with the rigid core of the molecule.

Taking into account the experimental results, both potentials $\Phi_s$ and $\Phi_{ij}$ are well described by an exponentially decaying functions [25,26]

$$\frac{\Phi_s}{k_B T} = -V_s \left[ \exp \left( -\frac{i}{\varsigma} \right) + \exp \left( -\frac{N+1-i}{\varsigma} \right) \right] \tag{6}$$

and

$$\frac{\Phi_{ij}}{k_B T} = -V_0 \frac{1}{|i-j|^2}. \tag{7}$$

Please note that both parameters $V_s$ and $V_0$ are positive, because all abovementioned interactions are attractive. For simplicity, we assume that the surface potential $\Phi_s$ [Equation (6)] is symmetric with respect to distances from surfaces. Furthermore, the characteristic length scale $\varsigma$, specifying the range of the surface interactions, has been chosen equal to one [27]. It should be pointed out that the inverse square distance dependence for the pair interlayer potential $\Phi_{ij}$ is in accordance with the theoretical result obtained for a pair of interacting surfaces [39,40]. Here, for convenience, the inverse reduced temperature $V_0 \sim 1/T$ has been used rather than the pure temperature [25,26].

The set of the OPs $\eta_i$ and $\sigma_i$, corresponding to the $i$th layer of the film composed of a stack of $N$ layers can be obtained by solving the system of $2N$ nonlinear self-consistent Equations (1)–(5), at a given number of film layers $N$, temperature $T$, and two parameters $\alpha$ and $V_s/V_0$ of the model. The distributions of the OPs $\eta_i$ and $\sigma_i$ across the $N = 100$ layer smectic film, at nine values of $V_0$ is shown in Figures 2a–d, 3e–h, and 4i [41], respectively.

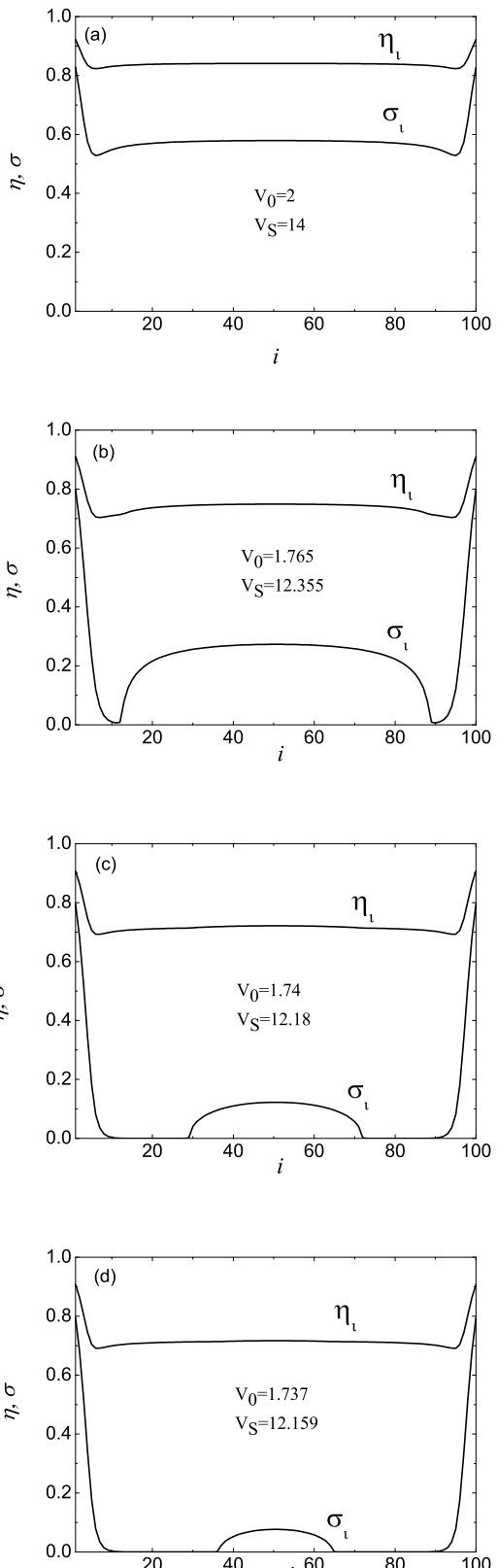

**Figure 2.** The set of thelocal $\eta_i$ and $\sigma_i$ OPs vs. $i$ [41]. In all cases, $V_s > 7V_0$.

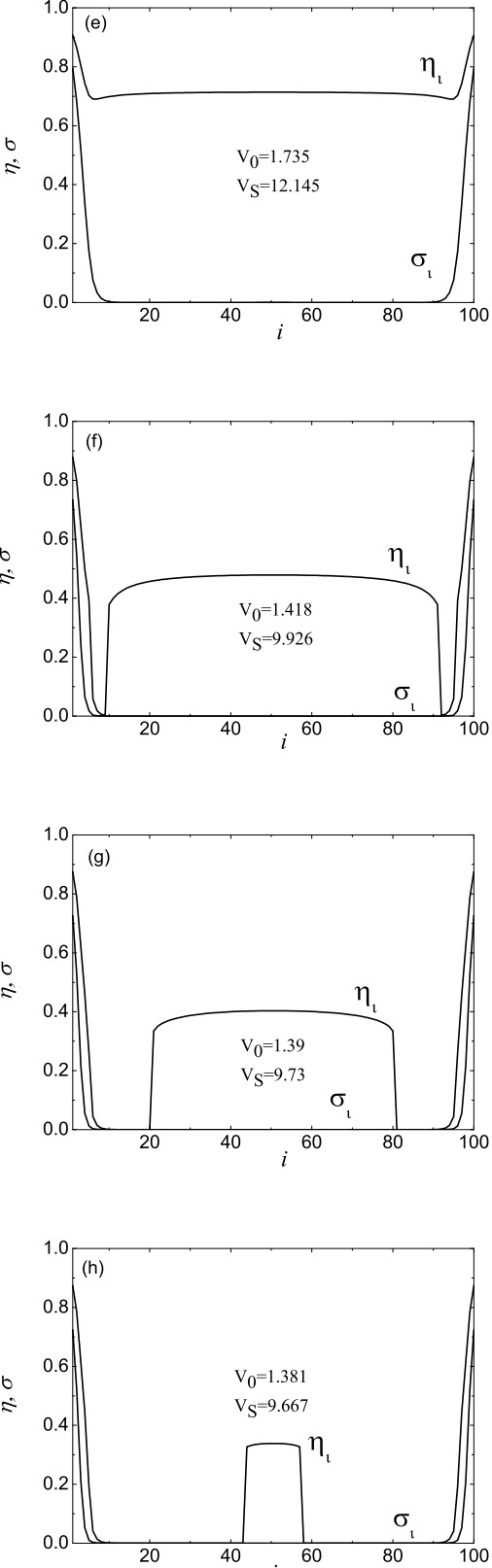

**Figure 3.** Same as in Figure 2a–d.

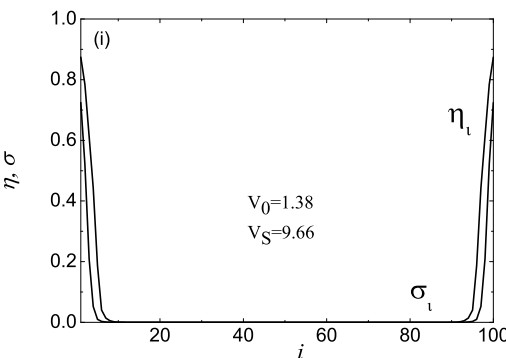

**Figure 4.** Same as in Figures 2a–d and 3e–h.

It should be pointed out that the ratio $V_s/V_0 > 7$ corresponds to the case of rather strong surface interactions.

"Calculations showed that both OPs $\eta_i$ and $\sigma_i$ are positive for all layers, when the temperature takes sufficiently low values ($V_0 \geq 1.765$) [41] (Figure 2a). This means that the SmA phase exists in the whole confined LC system. Because both the surface and interlayer interactions are attractive, smectic layers are more strongly stabilized in the middle part of the LC system than in the vicinity of the bounding surfaces. On the other hand, the studied system is surface stabilized, and hence the LC molecules exhibit the SmA ordering in domains close to surfaces. As the temperature further rises, the smectic ordering begins to vanish in the vicinity of the bounding surfaces and, simultaneously, the nematic ordering starts to arise in these domains [42]. This is illustrated in Figure 2b, where the orientational OP $\eta_i > 0$, whereas the translational OP $\sigma_i$ is equal to zero. Calculations showed that for $N = 100$, the nematic phase starts to arise at the temperature $T_1$ [41], corresponding to $V_0 = 1.765$, whereas the smectic ordering still prevails in the central domain of the LC system. Figure 2c,d shows how with further increase of temperature, the layer melting transition from the smectic-A to the nematic ordering propagates, largely into interior of the LC system. In turn, as shown in Figure 3e, at higher temperature $T_2$ (corresponding to $V_0 = 1.735$), the smectic phase completely disappears in the central domain of the LC system and, afterwards, for the temperature $T_3$ (corresponding to $V_0 = 1.418$), the local isotropic domain (associated with $\eta_i = 0$ and $\sigma_i = 0$) begins to form in the vicinity of the bounding surfaces, as shown in Figure 3f. As a consequence, Figure 4i shows that with increase of temperature, the frontiers between the isotropic and centrally arising nematic domains move (Figure 3g–h), until the nematic ordering completely vanishes (at the temperature $T_4$, corresponding to $V_0 = 1.38$). In the case when $V_0 \leq 1.38$, i.e., above the reduced temperature $T_4$, the isotropic phase occurs in whole the LC sample, except for the small domains close to surfaces, where the surface interactions hamper the disorder process, promoting the smectic order, which persists also at high enough temperatures. Accordingly, smectic layers formed in the immediate vicinity of each of surfaces can coexist with nematic and centrally formed smectic domain (Figure 2b–d), or can coexist with isotropic and centrally formed nematic domain (Figure 3f–h). When temperature increases, fronts between nematic or isotropic domains and the SmA domain, as well as fronts between isotropic and nematic domains move, mainly towards the center of the LC system. Results presented in Figures 2–4 show the very complex behavior of both orientational and translational OPs, due to the interplay between pair long-range intermolecular and nonlocal, relatively short-range [41] surface interactions. Calculations also showed that the SmA, nematic and isotropic phases can coexist, whereas the phase transitions from SmA to nematic, as well as from nematic to isotropic phases, as the temperature increases, does not occur simultaneously in the whole volume of the LC system but only in some domains of the LC sample. It should be pointed out that there are four characteristic temperatures, $T_i$, ($i = 1, 2, 3, 4$), at which particular phases arise or vanish. (Please note that $T_i \sim 1/V_0^{(i)}$ is the corresponding value of the reduced temperature.) For instance, at temperature $T_1$ the nematic phase starts to form in the vicinity of the bounding surfaces. Simultaneously, the smectic-A phase disappears

within these domains, as shown in Figure 2b. How it is shown in Figure 3e, the vanishing process of the smectic phase in the central domain of the LC system takes place at somewhat higher temperature $T_2$. In turn, Figure 3f shows that like the nematic phase, the isotropic phase begins to appear also in the vicinity of the bounding surfaces, but at temperature $T_3 > T_2$. Finally, at temperature $T_4 > T_3$, the nematic phase completely disappears." [42].

Clearly, when the system is not very thin, its interior (sufficiently far from surfaces) is controlled by interlayer interactions. However, when the thickness of a system is relatively small, in comparison with the range of surface interactions, the behavior of the system is dominated by surface anchoring couplings. Calculations showed [25] that the profiles presented in Figure 5 are qualitatively consistent with those derived also for $N = 25$, but assuming that surface potentials are strictly local and that two-layer potentials are independent of distance between the layers [7,19–21]. This indicates that the underlying method based on averaging such potentials at each iteration of self-consistent procedure applies for rather very thin real systems, entirely or almost entirely governed by surface anchoring interactions.

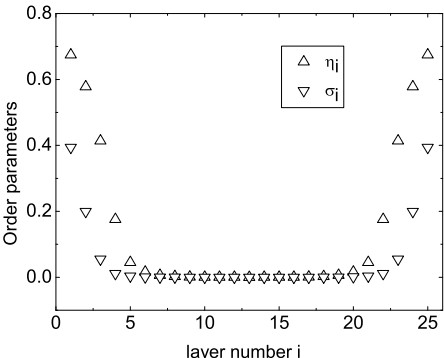

**Figure 5.** The set of the local $\eta_i$ and $\sigma_i$ OPs vs. $i$ [25] for $N = 25$, $V_S = 3V_0$, and $V_0 = 1.4$.

It should be pointed out that in the framework of the abovementioned mean-field approach for description of LC system confined in the microsized volume [25,26], the reduced temperature $T$ was defined as $T \sim 1/V_0$. In the case where it is necessary to calculate the temperature values with high accuracy, for example, in the case of unusual layer-thinning transition observed in FSSF, composed of partially fluorinated H10F5MOPP molecules [1], a precise definition of the dimensionless temperature is needed. It will be done in Section 2.2.3.

In turn, a new type of scaling behavior of the LC system interacting with the solid substrate will be analyzed in the next Section.

### 2.2.2. Finite-Size Effect in Thin LC Films on a  Solid Surface

Effects of surface ordering in LC systems confined between solid boundaries are of great theoretical and experimental interest.  In the previous Section a new theoretical approach for analyzing the effect of surfaces on local molecular ordering in thin LC systems with planar geometry of the smectic layers [25,26,41] was introduced.  These results showed that due to the interplay between pair long-range intermolecular forces and nonlocal, relatively short-range, surface interactions, both orientational and translational orders of molecules across confining cells are complex. In particular, it has been demonstrated that the SmA, nematic, and isotropic phases can coexist [25,26]. The phase transitions from SmA to nematic, as well as from nematic to isotropic phases, occur not simultaneously in the whole volume of the system but begin to appear locally in some domains of the LC sample. Phase-transition temperatures are demonstrated to be strongly affected by the thickness of the LC system. The dependence of the corresponding shifts of phase-transition temperatures on the layer number is shown to exhibit a power-law character. This new type of scaling behavior is concerned with the coexistence of local phases in finite systems. The influence of a specific character of

interactions of molecules with surfaces and other molecules on values of the resulting critical exponents now will be analyzed.

The set of the temperature parameter shifts $t_i(N) = V_0^{(i)}(N) - V_0^{(i)}(500)$, $i = 1, 2, 3, 4$ as a function of the number $N$ of smectic layers can be obtained by solving the system of $2N$ nonlinear self-consistent Equations (1)–(5), at a given number of film layers $N$, temperature $T$, and two parameters $\alpha$ and $V_s/V_0$ of the model, and results of calculation of the temperature parameter shift $t_i(N)$ (i=1,2,3,4) [41] is shown in Figure 6.

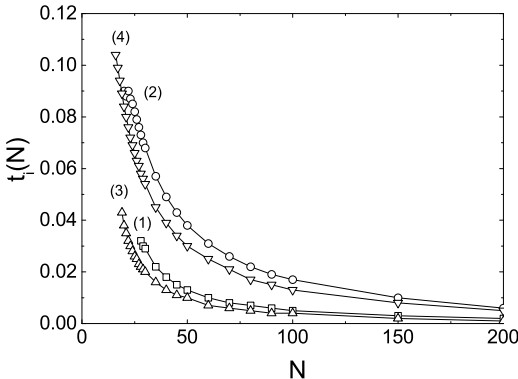

**Figure 6.** Temperature parameter shifts $t_1(N)$ (curve 1), $t_2(N)$ (curve 2), $t_3(N)$ (curve 3), $t_4(N)$ (curve 4) vs. the number $N$ of smectic layers [41].

Here $V_0^{(i)}(N)$ is the value of parameter $V_0^{(i)}$ determined for the system of the size $N$. It should be pointed out that values of $V_0^{(i)}(N = 500)$ are treated as the bulk values. The log-log plots (Figure 7) of the functions $t_i(N)$, $(i = 1, 2, 3, 4)$ distinctly showed that the temperature shifts undergo the scaling law [41]

$$t_i(N) \sim N^{\alpha_i}, \tag{8}$$

whereas the values of critical exponents $\alpha_i$ $(i = 1, 2, 3, 4)$ are being independent of $N$. Values of $\alpha_i$ $(i = 1, 2, 3, 4)$ corresponding to the appropriate local phase transitions are: $\alpha_1 = -1.402$, $\alpha_2 = -1.122$, $\alpha_3 = -1.488$, and $\alpha_4 = -1.172$, respectively. Calculations showed that the following approximate relations $\alpha_1 \approx \alpha_3$ and $\alpha_2 \approx \alpha_4$ are satisfied [41]. Critical exponents $\alpha_1$ and $\alpha_3$ describe processes of the appearance (at least, in the vicinity of the bounding surfaces) of the nematic and isotropic phases, respectively, while $\alpha_2$ and $\alpha_4$ characterize processes of disappearance (in the central domain of the LC system) of the SmA and nematic phases, respectively. Values of pairs of the exponents ($\alpha_1$, $\alpha_3$) and ($\alpha_2$, $\alpha_4$) distinctly differ each other because indices $\alpha_1$ and $\alpha_3$ correspond to the local phase transitions that take place in noncentral parts of the LC system, dominated by the surface interactions, while exponents $\alpha_2$ and $\alpha_4$ are associated with the local phase transitions within the central part of the LC system, dominated by the van der Waals interactions. Therefore, the values of critical exponents are determined by the type of interactions existing in the LC system [41]. Please note that the values of exponents $\alpha_1$ and $\alpha_3$ are not identical, as well as $\alpha_2$ is not exactly equal to $\alpha_4$, although exponents of a given pair ($\alpha_1$, $\alpha_3$) or ($\alpha_2$, $\alpha_4$) are associated with the same domain of the LC cell (i.e., peripheral or central regions, respectively). It should be pointed out that surface interactions play dominating role in the vicinity of the bounding surfaces, whereas the central part of the LC system is dominated by the intermolecular interactions. However, each of the indices of a given pair refers to transitions between different phases, characterized by different thermodynamic behavior of the studied LC system, which is why indices of a given pair are not equal. The analysis of ordering in the studied surface stabilized LC systems has been carried out for the set of the ratios between values of surface and intermolecular interactions [41]. This ratio corresponds to temperatures of transitions between the studied local phases, differently located with respect to surfaces of the considered LC system. All phases and phase

transitions between them are characterized by the wide range of values of that interaction ratio. It would be interesting to compare theoretical and experimental results for phase-transition temperatures and for profiles of order parameters across the surface stabilized LC systems. However, till now, there are no experimental results explaining these problems.

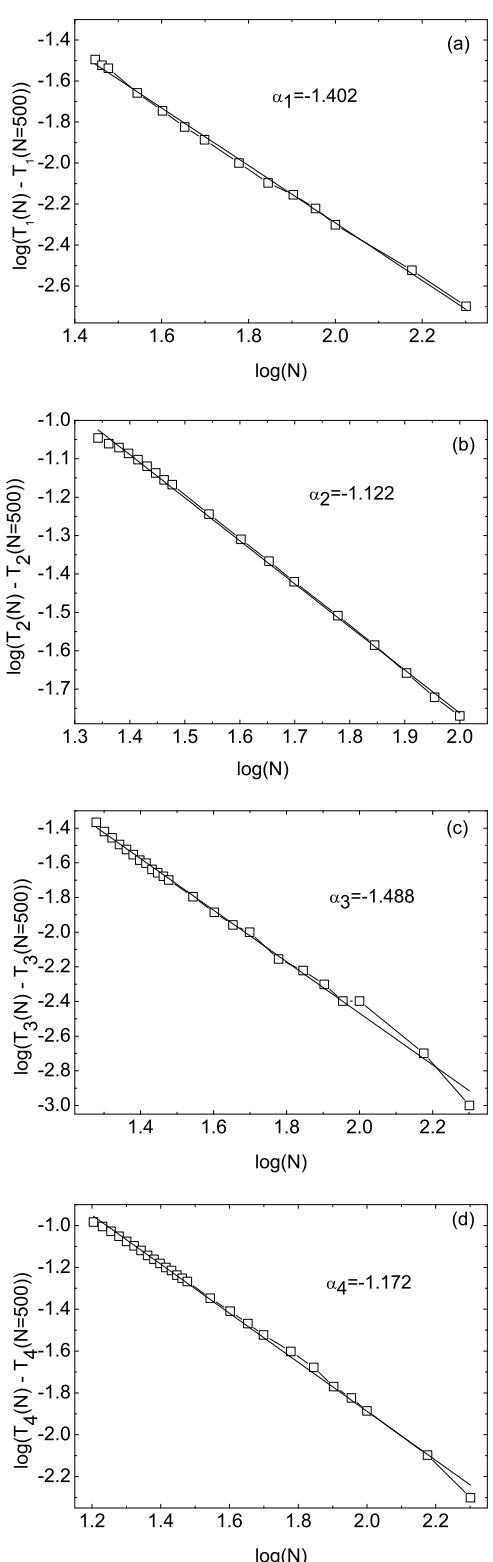

**Figure 7.** Values of critical exponents $\alpha_i$, $i = 1, 2, 3, 4$ [42] corresponding to the appropriate local phase transitions. The solid lines represent the best linear fits.

　　　The finite-size effect studied theoretically in this review can be a motivation for experimental studies, by applying, e.g., the fluorescence scanning laser confocal microscopy or the X-ray technique. The initial experimental results obtained by applying the technique of fluorescence scanning laser confocal microscopy [24,42,43] confirmed the inhomogeneity of molecular ordering in finite surface stabilized LC systems [42].

　　　It should be pointed out that the full phase diagram and the set of phase transitions of LC system interacting with the solid surface can be obtained in the framework of the mean-field approach which takes into account the translational-translational, orientational-orientational and mixed correlations [44,45]. In the next Section, it will be demonstrated by an example of a freely suspended smectic films.

### 2.2.3. Mean-Field Theory with Anisotropic Forces for Description of the Layer-Thinning Transition in FSSFs

　　　In this Section we will present an overview of mean-field approaches for describing the structural and thermodynamic properties, such as the Helmholtz free energy, entropy, and heat capacity, of free-standing smectic films. This will be done in the framework of the mean-field approaches, with anisotropic forces [7,12,17], where a free-standing smectic-A film is composed of $N$ discrete smectic layers with a thickness of the order of the molecular length $d$ and with total number of particles $M = \sum_{i=1}^{N} N_i$, where $N_i$ is the number of molecules per layer, which is assumed to be the same for all layers. The molecules within each layer are assumed to interact only with molecules of the same layer and those of the two neighboring ones. In the framework of these mean-field approaches, the set of potentials $\Phi_i$ $(i = 1, ..., N)$ within the $i$th smectic layer can be introduced [7,38]

$$\Phi_1(z_1, \beta_1) = -\frac{V_0}{3}\left[\frac{W_0}{V_0}\eta_1 + \eta_2 + \alpha\cos(2\pi z_1)\left(\frac{W_0}{V_0}\sigma_1 + \sigma_2\right)\right]P_2(\cos\beta_1),$$

$$\Phi_{1<i<N}(z_i, \beta_i) = -\frac{V_0}{3}\left[\sum_{j=i-1}^{i+1}\eta_j + \alpha\cos(2\pi z_i)\sum_{j=i-1}^{i+1}\sigma_j\right]P_2(\cos\beta_i),$$

$$\Phi_N(z_N, \beta_N) = -\frac{V_0}{3}\left[\frac{W_0}{V_0}\eta_N + \eta_{N-1} + \alpha\cos(2\pi z_N)\left(\frac{W_0}{V_0}\sigma_N + \sigma_{N-1}\right)\right]P_2(\cos\beta_N), \tag{9}$$

where $z_i$ is the dimensionless distance through the smectic film, $V_0$ is the force potentials which is responsible for the molecule-molecule interaction, $W_0$ is the parameter corresponding to "enhanced" pair interactions in the bounding layers, and the constant $\alpha$ implicitly characterizes molecular packing across smectic layers [38]. Physically, these approaches indicate that we replace $V_0$ by $W_0$ within the first and last layers, whereas for all interior layers $1 < i < N$ the interaction coefficient $V_0$ has not been changed. It should be pointed out that the effective anisotropic potential $\Phi_i$ $(i = 1, ..., N)$ in the form of Equation (9) is a reduced version of the potential $\Phi_i$ in the form of Equation (1).

　　　The set of OPs $\eta_i$ and $\sigma_i$ corresponding to the $i$th layer of the smectic film composed of a stack of $N$ SmA layers in air can be obtained by solving the system of $2N$ nonlinear self-consistent Equations (3)–(5), with the effective anisotropic potential $\Phi_i$ $(i = 1, ..., N)$ in the form of Equation (9), at a given number of film layers $N$, temperature $T$, and the two parameters $\alpha$ and $W_0/V_0$ of the model. Having obtained the set of OPs $\eta_i$ and $\sigma_i$ $(i = 1, ..., N)$, one can calculate the full Helmholtz free energy of the LC system as $F(N, T) = \frac{1}{N}\sum_{i=1}^{N} F_i$, where $F_i$ is the Helmholtz free energies corresponding to the $i$th layer. In turn, the dimensionless full Helmholtz free energy per molecule for each layer can be

written as $f = \frac{1}{N} \sum_{i=1}^{N} f_i$, where $f_i = \frac{F_i}{N_i V_0}$ is the dimensionless Helmholtz free energy corresponding to the $i$th layer, which can be calculated as [7,12,17,46]

$$f_1 = \frac{1}{6} \left[ \frac{W_0}{V_0} \eta_1 \left( \frac{W_0}{V_0} \eta_1 + \eta_2 \right) + \alpha \frac{W_0}{V_0} \sigma_1 \left( \frac{W_0}{V_0} \sigma_1 + \sigma_2 \right) - 2\theta \ln Q_1 \right],$$

$$f_{1<i<N} = \frac{1}{6} \left[ \eta_i \sum_{j=i-1}^{i+1} \eta_j + \alpha \sigma_i \sum_{j=i-1}^{i+1} \sigma_j - 2\theta \ln Q_i \right],$$

$$f_N = \frac{1}{6} \left[ \frac{W_0}{V_0} \eta_N \left( \frac{W_0}{V_0} \eta_N + \eta_{N-1} \right) + \alpha \frac{W_0}{V_0} \sigma_N \left( \frac{W_0}{V_0} \sigma_N + \sigma_{N-1} \right) - 2\theta \ln Q_N \right], \tag{10}$$

where $\theta = \frac{3k_B T}{V_0}$ is the dimensionless temperature and $Q_i = \int_{(i-1)}^{i} dz \int_0^1 h_i(x,z) dx$, $(i = 1, ..., N)$ is the partition function of the $i$th layer, respectively.

In the scope of our research interest is also to investigate the experimentally observed phenomenon of the stepwise reduction of the value of heat capacity [7,17]

$$c_v = \frac{C_v}{M k_B} = \frac{T}{M k_B} \left( \frac{\partial S}{\partial T} \right)_v = \theta \left( \frac{\partial s}{\partial \theta} \right)_v \tag{11}$$

as the temperature $\theta$ is raised above the dimensionless $\theta_{AI}(\text{bulk}) = 3k_B T_{AI}(\text{bulk})/V_0$ [1]. In order to calculate the values of $c_v$, one must first calculate the entropy of the system per molecule $s = \frac{1}{N} \sum_{i=1}^{N} s_i$, where

$$-\frac{\theta}{3} s_i = 2 f_i + \frac{\theta}{3} \ln Q_i, \ (1 \leq i \leq N). \tag{12}$$

Here $s_i = \frac{S_i}{N_i k_B}$ is the dimensionless entropy per molecule corresponding to the $i$th layer.

Recently, an experimental phenomenon of the stepwise behavior of surface tension upon heating the smectic-A film above $\theta_{AI}(\text{bulk})$ has been observed [9]. It was shown that the film tension $\Gamma$, at each thinning abruptly jumps to a lower value and then continues to increase with a smaller slope [9]. In the framework of the mean-field approach, the dimensionless surface tension $\gamma = \Gamma \frac{a}{V_0}$ of the smectic film per molecule at constant volume $v = V/M$, can be calculated as [11,46]

$$\gamma = \left( \frac{\partial f}{\partial \bar{a}} \right)_{p,v,\theta} = - \left( z \frac{\partial f}{\partial z} \right)_{p,v,\theta}, \tag{13}$$

where $p$ is the dimensionless pressure per molecule and $a = \bar{a} d^3$ is the area per molecule at constant $p$ and $v$. At the same time, the calculation of surface tension $\Gamma = \left( \frac{\partial F}{\partial A} \right)_{P,V,T} = -\frac{1}{A} \left( z \frac{\partial F}{\partial z} \right)_{P,V,T}$ took into account the fact that $V = ANd = \text{const}$, where $F$ is the Helmholtz free energy of the smectic film and $A$ is the LC/vacuum interface area.

In the case when the FSSF is subjected to the external electric field **E** directed both across $\left( \mathbf{E} \parallel \hat{\mathbf{k}} \right)$ and along $\left( \mathbf{E} \parallel \hat{\mathbf{i}} \right)$ the smectic layers, the set of effective anisotropic potentials $\Phi_i$ $(i = 1, ..., N)$ can be rewritten in the form [11]

$$\Phi_1\left(z_1,\beta_1\right) =$$
$$-\frac{V_0}{3}\left[\frac{W_0}{V_0}\eta_1 + \eta_2 + \Delta\mathcal{C}\left(\eta_1\right) + \alpha\cos\left(2\pi z_1\right)\left(\frac{W_0}{V_0}\sigma_1 + \sigma_2\right)\right]P_2(\cos\beta_1),$$

$$\Phi_{1<i<N}\left(z_i,\beta_i\right) =$$
$$-\frac{V_0}{3}\left[\sum_{j=i-1}^{i+1}\eta_j + \Delta\mathcal{C}\left(\eta_i\right) + \alpha\cos\left(2\pi z_i\right)\sum_{j=i-1}^{i+1}\sigma_j\right]P_2(\cos\beta_i),$$

$$\Phi_N\left(z_N,\beta_N\right) =$$
$$-\frac{V_0}{3}\left[\frac{W_0}{V_0}\eta_N + \eta_{N-1} + \Delta\mathcal{C}\left(\eta_N\right) + \alpha\cos\left(2\pi z_N\right)\left(\frac{W_0}{V_0}\sigma_N + \sigma_{N-1}\right)\right]P_2(\cos\beta_N), \tag{14}$$

where

$$\mathcal{C}\left(\eta_i\right) = \begin{cases} \eta_i + \frac{1}{2}, & \text{for } \mathbf{E}\parallel\hat{\mathbf{k}}, \\ 1 - \eta_i, & \text{for } \mathbf{E}\parallel\hat{\mathbf{i}}, \end{cases}$$

and $\Delta = \frac{\epsilon_0\epsilon_a E^2}{n_0 V_0}$ is the dimensionless parameter corresponding to the electric field $\mathbf{E}$ applied across or along the smectic layers. Here $\epsilon_0$ is the dielectric permittivity of vacuum, $\epsilon_a$ is the dielectric constant of the smectic film, and $n_0$ is the number of density. In the framework of the mean-field approach, the dimensionless Helmholtz free energy corresponding to the $i$th layer can be written as [11]

$$f_1 = \frac{1}{6}\left[\frac{\overline{W_0}}{V_0}\eta_1\left(\frac{\overline{W_0}}{V_0}\eta_1 + \eta_2\right) + \alpha\frac{W_0}{V_0}\sigma_1\left(\frac{W_0}{V_0}\sigma_1 + \sigma_2\right) - 2\theta\ln Q_1\right],$$

$$f_{1<i<N} = \frac{1}{6}\left[\eta_i\left(\Delta+1\right)\sum_{j=i-1}^{i+1}\eta_j + \alpha\sigma_i\sum_{j=i-1}^{i+1}\sigma_j - 2\theta\ln Q_i\right],$$

$$f_N = \frac{1}{6}\left[\frac{\overline{W_0}}{V_0}\eta_N\left(\frac{\overline{W_0}}{V_0}\eta_N + q_{N-1}\right) + \alpha\frac{W_0}{V_0}\sigma_N\left(\frac{W_0}{V_0}\sigma_N + \sigma_{N-1}\right) - 2\theta\ln Q_N\right], \tag{15}$$

where $\frac{\overline{W_0}}{V_0} = \frac{W_0}{V_0} + \Delta$.

Equations from (3) to (5), with the effective anisotropic potential $\Phi_i$ $(i = 1,...,N)$ in the form of Equations (9)–(12) and (15) are the relations which are needed to calculate both the structural, optical and thermodynamic properties of the free-standing SmA films. The set of external parameters used in calculations are $N$, $\alpha$, and $W_0/V_0$, respectively. For the case of films composed of the partially fluorinated H10F5MOPP molecules, both calorimetric and optical reflectivity studies were carried out with initially 25-layer thick films, above the bulk SmA-Isotropic transition temperature ($T_{AI}$(bulk) $\sim$ 358 K). Taking into account this fact, in the theoretical investigations the initial thickness of the film was chosen as being equal to $N = 25$ [7,11,12,17,19–21]. According to the McMillan's theory [38], the first-order bulk $AI$ transition occurs for $\alpha \geq 0.98$, so, the choosing of $\alpha = 1.05$ is acceptable. When choosing the value of $W_0/V_0$, one is usually guided by the fact that the partially fluorinated free-standing smectic films composed of the H10F5MOPP molecules are stable above the $\theta_{AI}$(bulk). This allows the assumption that the value of the interaction constant $W_0$ should be greater than $V_0$. In the number of theoretical investigations [7,11,12,17,19–21] the strong surface-enhanced pair interactions with $W_0 = 5\,V_0$ has been chosen. Taking into account that the partially fluorinated compound H10F5MOPP has bulk SmA-I transition temperature $T_{AI}$(bulk) $\sim$ 358 K, ($\theta_{AI}$(bulk) $\sim$ 0.675) and for $\alpha = 1.05$, according to the McMillan's theory [38], the value of $k_B T_{AI}$(bulk)$/0.2202 V_0 = 1.021$, one can estimate that the value of $V_0$ is equal to $\sim 2.2 \times 10^{-20}$ J. Please note that values of the dimensionless temperature $\theta = \frac{3k_B T}{V_0}$ often vary between 0.60 ($\sim$ 318.2 K) and 0.80 ($\sim$ 424.3 K) [7,11,12,17].

In the next Section we will review several examples of numerical simulation of the layer-thinning transitions in free-standing partially fluorinated smectic films as the temperature is increased above $T_{AI}$(bulk).

### 2.2.4. Layer-Thinning Transitions in Free-Standing Partially Fluorinated Smectic Films

"When the temperature $\theta$ is slowly increased above $\theta_{AI}$(bulk) towards either the nematic or isotropic phases, competition between surface and finite-size effects leads to unusual properties of FSSFs. High-resolution optical reflectivity investigations show [33] that in the partially fluorinated compound, such as 2-4-(1,1-*dihydro-2-(2-perfluorobutoxy) perfluoroethoxy) phenyl-5-octyl pyrimidine* (H8F(4,2,1)MOPP), the order of the surface layers appears to enhance, so they become ordered at temperatures well above $\theta_{AI}$(bulk). The temperature $\theta'$s effect on the behavior of the orientational $\eta_i(\theta)$ and translational $\sigma_i(\theta)$ OPs in the smectic film with $N = 25$ layers has been investigated numerically by solving the set of $2N$ self-consistent nonlinear equations from (3) to (5), with the effective anisotropic potential $\Phi_i$ ($i = 1, ..., N$) in the form of Equation (9) [7], and the numerical result is shown in Figure 8a,b.

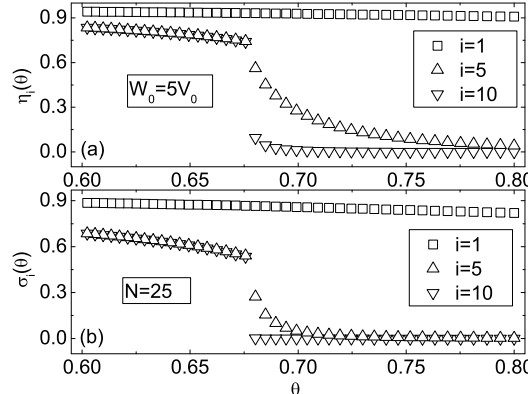

**Figure 8.** Plot of $\eta_i(\theta)$ ($i = 1, 5, 10$) (**a**) and $\sigma_i(\theta)$ ($i = 1, 5, 10$) (**b**) vs. $\theta$ [7].

The set of the model parameters used in these calculations are $N = 25$, $\alpha = 1.05$, and $W_0 = 5 V_0$, respectively. In the low-temperature region $0.60 \leq \theta \leq 0.675$ (318.2 K $\leq T \leq$ 358 K), results for orientational $\eta_i(\theta)$ (Figure 8a and translational $\sigma_i(\theta)$ OPs (Figure 8b OPs showed [7] that these equations have a stable unique solution, which is characterized by high values of $\eta_i(\theta)$ (Figure 8a, squares and up and down triangles) and $\sigma_i(\theta)$ OPs (Figure 8b, squares and up and down triangles), both in the vicinity of the bounding surfaces, as well as near the film center. In the high-temperature region $0.685 \leq \theta \leq 0.8$ (363.3 K $\leq T \leq$ 424.3 K), one also has a stable unique solution, which is characterized by vanishing both OPs $\eta_i(\theta)$ and $\sigma_i(\theta)$ near the film center, whereas in the vicinity of the bounding surfaces, both OPs still maintain relatively high values. In [7,19], this type of solution was called a "quasi-smectic" state. At intermediate temperatures $0.675 \leq \theta \leq 0.685$ (358 K $\leq T \leq$ 363.3 K) both types of solutions of the self-consistent equations exist, although, for clarity, Figure 8 shows only the quasi-smectic profiles.

Calculations also showed that both $\eta_i(\theta)$ and $\sigma_i(\theta)$ profiles demonstrate strong ordering in the vicinity of the bounding surfaces, due to the stronger pair interactions within the first and last layers than for all interior layers, which decreases rapidly with distance from those surfaces. For instance, both the $\eta_i(\theta)$ (Figure 8a, up triangles ($i = 5$)) and $\sigma_i(\theta)$ (Figure 8b, up triangles ($i = 5$)) OPs fall continuously to some finite values [7], whereas those parameters corresponding to the interior layers close to the film center (Figure 8a,b, down triangles ($i = 10$)) drop to 0.

Furthermore, on the basis of the behavior of the free energy, one can calculate the values of the layer-thinning transition temperatures [7]. For instance, in the case of strong ($W_0 = 5 V_0$) "enhanced" pair interactions in the bounding layers, the value of the temperature $\theta_{AI}(N = 25)$ is equal to

$\sim 0.678$ ($T_{AI}(N = 25) \sim 359.6\ K$). Here $\theta_{AI}(N = 25)$ and $T_{AI}(N = 25)$ denote the dimensionless and dimensional layer-thinning transition temperatures, respectively. According to these calculations [7], the distributions of the OPs $\eta_i(\theta)$ and $\sigma_i(\theta)$ across the 25-layer smectic film, at three dimensionless temperatures $\theta = 0.65$ ($\sim 344.74\ K$), 0.67 ($\sim 355.35\ K$), and 0.69 ($\sim 366\ K$), are characterized by a monotonic decrease of both $\eta_i(\theta)$ and $\sigma_i(\theta)$ with increasing distance (or number of layers) from the bounding surface towards the interior of the film.

In the case of strong ($W_0 = 5\ V_0$) "enhanced" pair interactions in the bounding layers, (see Figure 9a–c [7]) these distributions are characterized by minima in the middle part of the film and decreasing values of these OPs with increase in temperature.

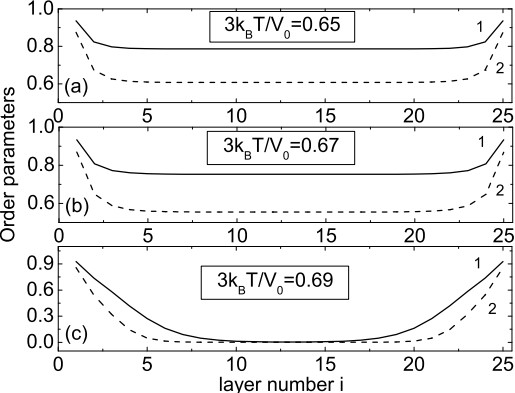

**Figure 9.** Plot of $\eta_i$ (curves 1) and $\sigma_i$ (curves 2) vs. *i* [7] for three temperature values $\theta$: (**a**) 0.65, (**b**) 0.67, and (**c**) 0.69, respectively.

Having obtained the profiles of OPs $\eta_i(\theta)$ and $\sigma_i(\theta)$, and using Equations (10) and (11), the distributions of both the dimensionless Helmholtz free energy $f(i)$ (Figure 10a) and entropy $s(i)$ (Figure 10b) can be calculated [7].

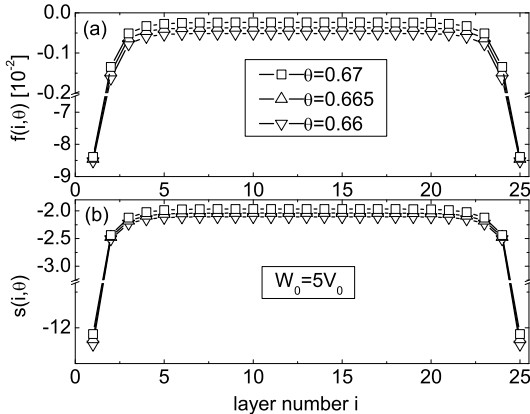

**Figure 10.** Plot of the Helmholtz free energy $f(i)$ (**a**) and entropy $s(i)$ (**b**) vs. *i* [7], for three values of $\theta$: 0.66 (down triangles), 0.665 (up triangles), and 0.67 (squares), respectively.

Calculations that were performed for three temperature $\theta$ values [7]: 0.66 (down triangles), 0.665 (up triangles), and 0.67 (squares), showed that the free-energy profiles demonstrate monotonic growth of the value of $f(i)$ up to the 8th layer from each boundary, where the function $f(i)$ saturates and does not change with further increase of *i*. Physically, this means that all film layers are subjected to attractive forces from the bounding surfaces. The results of calculations shown in Figure 8 indicate that at temperatures close to the layer-thinning value $\theta_{AI}(N = 25) \sim 0.678$, strong ordering takes place only in the vicinity of the bounding surfaces, whereas far from the surfaces ordering drops to lower

values than in the bounding layers. As a result, anyone can find that when the temperature varies from below $\theta_{AI}(N = 25)$ to a lower value $\theta = 0.66$, there are smaller differences between the Helmholtz free-energy $f(i)$ profiles (see Figure 10a, contrasting the up and down triangles from the squares). The same tendency can be seen in the case of the entropy $s(i)$ profiles (see Figure 10b). The distribution of the free-energy profiles across the smectic film changes dramatically as the temperature increases. When the layer-thinning transition temperature corresponding to the case of strong interaction with $W_0 = 5\,V_0$ ($\theta_{AI}(N = 25) \sim 0.678$ ($T_{AI}(N = 25) \sim 359.6$ K)) for a film initially containing 25 layers is reached, the interior layers become unstable and the system undergoes the discontinuous transition to the quasi-smectic state [7]. Such an effect has been seen earlier in the behavior of the order parameters in Figure 8. The distributions of both the $f(i)$ and $s(i)$ profiles in the high-temperature region $\theta > 0.678$ are shown in Figure 11a,b, for temperatures $\theta = 0.68$ (squares), 0.685 (up triangles), and 0.69 (down triangles), respectively.

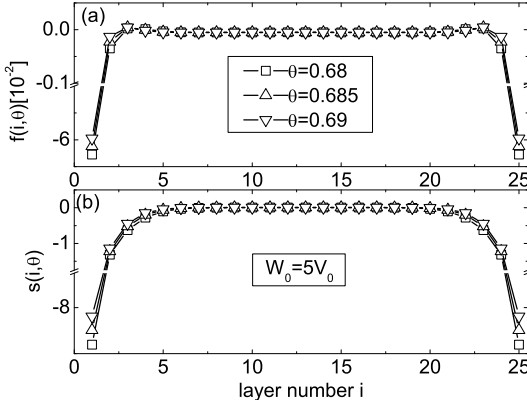

**Figure 11.** Same as in Figure 10a,b, but for next three temperature $\theta$ values [7]: 0.68 (squares), 0.685 (up triangles), and 0.69 (down triangles), respectively.

Now the distribution of $f(i)$ across the $N = 25$ layer smectic film is characterized by maxima in the vicinity of both bounding surfaces. Calculations showed that the forces acting on the interior layers are in the opposite direction to the attractive ones. As a result, the interior layers are compressed and squeezed by the bounding layers.

The temperature $\theta'$s effect on the Helmholtz free energy $f(\theta)$ and entropy $s(\theta)$ of the smectic films containing 25, 13, 11, 10, 8, and 6 layers are shown in Figures 12a,b and 13a,b, respectively.

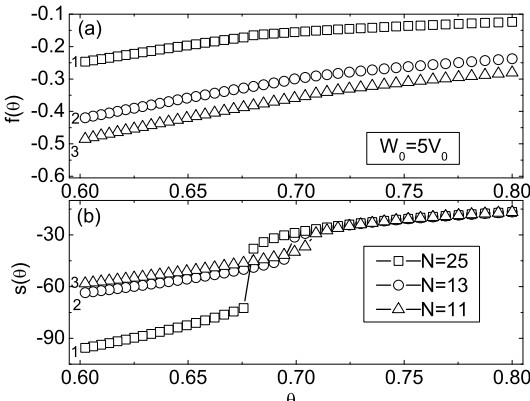

**Figure 12.** (a) Plot of $f(\theta)$ vs. $\theta$ [7], for three film thicknesses: $N = 25$ (curve 1),13 (curve 2), and 11 (curve 3), respectively. (b) Same as (a), but for $s(\theta)$.

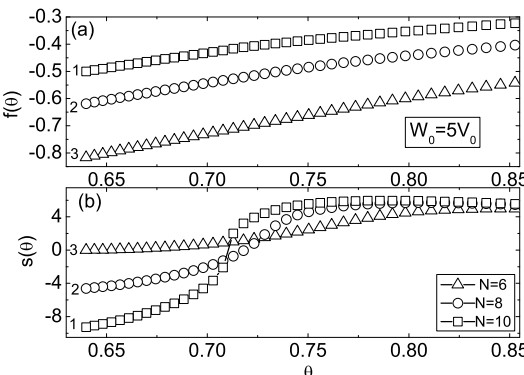

**Figure 13.** (**a**,**b**). Same as in Figure 12a,b [7], but for next three film thicknesses: $N = 10$ (curves 1), 8 (curves 2), and 6 (curves 3), respectively.

Calculations were carried out for the set of the model parameters [7] $\alpha = 1.05$ and $W_0 = 5\,V_0$, respectively. In Figure 12a,b the distribution of both the $f(\theta)$ and $s(\theta)$ vs. $\theta$, for several film thicknesses: $N = 25$ (curve 1), $N = 13$ (curve 2), and $N = 11$ (curve 3) (Figure 12a,b), and $N = 10$ (curve 1), $N = 8$ (curve 2), and $N = 6$ (curve 3) (Figure 13a,b), respectively are shown. Results of calculations showed that the SmA-I transition occurs through the sequence of layer-thinning transitions $25 \rightarrow 13 \rightarrow 11 \rightarrow 10...$, as the temperature is increased. The calculated free energy $f(\theta)$ per molecule for the 25-layer thick film vs. $\theta$ is shown in Figure 12a (curve 1), and demonstrates smooth behavior with increase of $\theta$, whereas the value of $s(\theta)$ (Figure 12b, (curve 1)) demonstrates a discontinuous rise at $\theta_{AI}(N = 25) \sim 0.678$ ($T_{AI}(N = 25) \sim 359.6$ K) greater than 40 $k_B$ per molecule, due to the transition to the quasi-smectic state and corresponding change in slope of the free-energy curves. A similar discontinuity in $s(\theta)$ is seen in Figure 13b for $N = 10$ (curve 1). Discontinuities in $s(\theta)$ also occur for the other values of $N$, but are not seen in the figures due to the fixed vertical length scale.

Following the transition of $N$-layer film to the quasi-smectic state, it has been determined the number of layers $(N - n)$ remaining in the film with non-vanishing smectic order near the film center to be such as to provide a lower free energy than the $N$-layer state at the same temperature, as well as with a higher transition temperature. Calculations showed [7] that the next stable state with lower free energy occurs at $N = 13$, then at $N = 11$, etc. The corresponding layer-thinning temperatures $\theta_{AI}(N)$ are: $\theta_{AI}(N = 25) \sim 0.678$ ($\sim 359.6$ K), $\theta_{AI}(N = 13) \sim 0.697$ ($\sim 369.7$ K), $\theta_{AI}(N = 11) \sim 0.706$ ($\sim 374.44$ K), $\theta_{AI}(N = 10) \sim 0.7106$ ($\sim 377$ K), $\theta_{AI}(N = 9) \sim 0.717$ ($\sim 380.3$ K), $\theta_{AI}(N = 8) \sim 0.729$ ($\sim 386.6$ K), $\theta_{AI}(N = 7) \sim 0.736$ ($\sim 390.3$ K), $\theta_{AI}(N = 6) \sim 0.743$ ($\sim 394$ K), etc.

In the following Section several structural, thermodynamic and optical properties of free-standing smectic films, as the temperature is above $\theta_{AI}(\text{bulk})$, will be considered. [7]

### 2.2.5. Heat Capacity, Surface Tension, Disjoning Pressure and Optical Refectivity of FSSFs

A great variety of thermodynamic properties has been observed in FSSFs. Among other, a very interesting phenomena is the stepwise reduction of heat capacity $c_v(\theta)$ when the temperature is increased above $\theta_{AI}(\text{bulk})$. In the framework of the abovementioned mean-field approach, the temperature $\theta$'s effect on the $c_v(\theta)$ at constant volume of the smectic film with 25 layers, in two cases of strong interactions $W_0 = 5\,V_0$ and 10 $V_0$, has been investigated numerically [7] and the results are shown in Figure 14a,b [7].

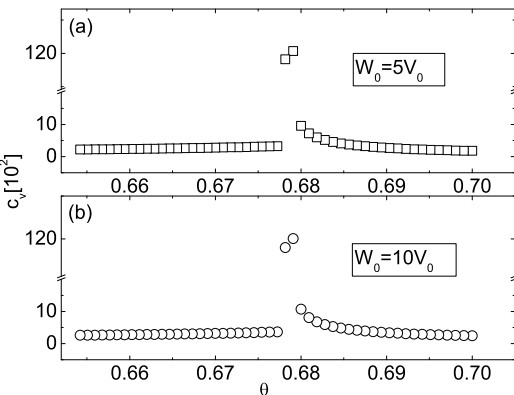

**Figure 14.** (**a**) Plot of $c_v(\theta)$ vs. $\theta$ [7]. Here the interaction constant $W_0 = 5.0 \ V_0$. (**b**) Same as (**a**), but $W_0 = 10 \ V_0$.

Calculations showed that the heat capacity $c_v(\theta) \sim 10^4$ anomaly (i.e., heat-capacity peaks) (Figure 14a) at temperature $\theta_{AI}(N = 25) \sim 0.678$ ($T_{AI}(N = 25) \sim 359.6$ K), is associated with the interior first-order SmA-I transition, where the entropy change is greater than $40 \ k_B$ (Figure 12b, curve 1) per molecule, and demonstrates a discontinuous rise at $\theta_{AI}(N = 25)$, whereas the value of $c_v(\theta)$ (Figure 14a), in the temperature range $0.655 \le \theta \le 0.677$, varies between 280, at $\theta \sim 0.655$, and 450, at $\theta = 0.677$, respectively. Please note that in the case when the "enhanced" pair interactions in the bounding layers are in two times stronger $W_0 = 10 \ V_0$, the temperature $\theta$'s effect on the $c_v(\theta)$ has the same qualitative behavior (see Figure 14b) and the value of the layer-thinning transition temperature $\theta_{AI}(N = 25)$ is practically the same as in the weaker case. In the temperature range $0.655 \le \theta \le 0.677$ the value of $c_v(\theta)$ (Figure 14b) varies between 300, at $\theta \sim 0.655$, and 480, at $\theta = 0.677$, respectively.

"Heat capacity $c_v(\theta)$ values of the partially fluorinated H10F5MOPP 25-layer film, calculated in the framework of the mean-field approach, at temperature $\theta \sim 0.674$ ($\sim 357$ K), below both the bulk SmA-Isotropic transition temperature and the layer-thinning transition temperature corresponding to strong ($W_0 = 5 \ V_0$) interactions in the bounding layers, are equal to $\sim 420$, or $\sim 82.5 \ \left[\frac{\mu J}{K \ cm^2}\right]$, or $\sim 5.74 \times 10^{-21} \ \left[\frac{J}{K \ mol}\right]$ [7], respectively. In turn, the measured, by means of calorimetric techniques, value of $C_P$, at the same temperature corresponding to "plateau" values of the heat capacity, is equal to $\sim 80 \ \left[\frac{\mu J}{cm^2 K}\right]$, or $\sim 5.9 \times 10^{-21} \ \left[\frac{J}{K \ mol}\right]$ [1]. Hence, it has been obtained a good agreement between the theoretically predicted [7] and experimentally obtained [1] results. In recalculations of the theoretical values of $c_v(N)$ per H10F5MOPP molecule to compare with the measured $C_p(N)$ values, it has been used the fact that the total number of molecules $M$ per unit area in the film, denoted as $n_s$, can be estimated as $n_s = n_0 l$, where $n_0 \sim 1.5 \times 10^{21} \ cm^{-3}$ is the number density and $l = Nd$ is the thickness of the $N$-layer film. Since $d$ is of the order of the molecular length $\sim 3.0$ nm [33], $n_s$ can be estimated as $n_s \sim N \times 4.5 \times 10^{14} cm^{-2}$.

The result of comparing of the calculated value on $c_v(N)$, obtained in the framework of the mean-field approach, and the experimentally measured values of $C_v(N)$ shows that the extended McMillan's approach "enhanced" by anisotropic interactions in the bounding layers, with $W_0 = 5 \ V_0$, is more suitable for describing both the structural and thermodynamic properties of a partially fluorinated H10F5MOPP smectic film than with $W_0 = 10 \ V_0$, which gives $c_v \sim 450$, or $\sim 86.7 \ \left[\frac{\mu J}{K \ cm^2}\right]$, at temperature $\theta \sim 0.674$ ($\sim 357$ K).

The calculated data [7] on the dimensionless heat capacity $c_v(N)$ per molecule, and the recalculated dimensional heat capacity $C_v(N)$, corresponding to $N$ layer films, as well as the "plateau" temperatures $\theta(N)$ for the sequence of the abovementioned layer-thinning transitions (with $W_0 = 5V_0$) are collected in Table 1. Calculations showed that these plateau temperatures satisfy [7]: $\theta(25) < 0.678 < \theta(13) < 0.697 < \theta(11) < 0.706 < \theta(10) < 0.7106 < \theta(9) < 0.717 < \theta(8) < 0.729 < \theta(7) < 0.736 < \theta(6) < 0.743$, where the numbers correspond to the successive layer-thinning transition temperatures given earlier. The observed data on $C_p(N)$ for the free-standing partially

fluorinated H10F5MOPP smectic films also correspond to a series of "plateau" values for the sequence of the layer-thinning transitions $25 \to 15 \to 11 \to 9 \to 8...$ etc., [1]. In the range of film thicknesses investigated, the reduction of $C_v(N)$ is, at least qualitatively, in agreement with the experimentally observed decrease of $C_p(N)$ with decrease of $N$.

**Table 1.** The calculated data on $c_v(N)$ and measured values of $C_v(N)$, corresponding to the sequence of the layer-thinning transition in free-standing partially fluorinated H10F5MOPP smectic films.

| | (Temp.) | (Theor.) | (Theor.) | | (Temp.) | (Exp.) |
|---|---|---|---|---|---|---|
| N(theor.) | $\theta(N)$(theor.) | $c_v(N)$ | $C_v(N)[\frac{\mu J}{cm^2 K}]$ | N(exp.) | $\theta(N)$(exp.) | $C_p(N)[\frac{\mu J}{cm^2 K}]$ |
| 25 | 0.674 | 420 | 82.5 | 25 | 0.674 | 80 |
| 13 | 0.685 | 264 | 52 | 15 | 0.676 | 48 |
| 11 | 0.702 | 215 | 42.4 | 11 | 0.678 | 35 |
| 10 | 0.708 | 184 | 36.3 | 9 | 0.68 | 30 |
| 9 | 0.711 | 155 | 31 | 8 | 0.682 | 25 |
| 8 | 0.722 | 134 | 27 | 7 | 0.684 | 22 |
| 7 | 0.73 | 117 | 23 | 6 | 0.686 | 18 |
| 6 | 0.741 | 85 | 17 | - | - | - |

Comparisons of the theoretical and experimental reductions of heat-capacity values at "plateau" regions in thin smectic films away from the layer-thinning transition temperatures should be unaffected by questions of the layer-thinning mechanisms. Nevertheless, these mechanisms may affect the "anomalies" shown by the heat-capacity peaks in Figure 14. Such anomalies have not been presented in experimental studies of SmA layer-thinning transitions, to our knowledge, but only in studies of SmA to hexatic-B transitions of smectic films [22], and we hope the present review will spur further experimental work in this direction [7].

Recently, it has been carried out the high-resolution study of the film tension $\gamma$ as the film is heated through the layer-by-layer melting process, and the sawtooth behavior of the surface tension upon heating the FSSF above $\theta_{AI}$(bulk) was observed [9,10]. The understanding of how confinement influences the $\gamma(\theta)$ of thin smectic film when one or several layer(s) is(are) squeezed-out to meniscus has been investigated theoretically [11,12,47]. It has been done in the framework of the mean-field approach with anisotropic forces [7]. "The temperature $\theta$'s effect both on the dimensionless Helmholtz free energy $f(\theta)$ (see Equation (10)) and surface tension $\gamma(\theta)$ (see Equation (13)) per H10F5MOPP molecule in the smectic film, in the case when there is no electric field $E = 0$ or $\Delta = 0$, corresponding to a sequence of layer-thinning transitions $25 \to 13 \to 11 \to 10 \to 9 \to 8 \to 7 \to 6$, has been investigated numerically by solving the set of $2N$ self-consistent nonlinear equations from (3) to (5), with the effective anisotropic potential $\Phi_i$ ($i = 1, ..., N$) in the form of Equation (9). The results are shown in Figure 15a,b [11].

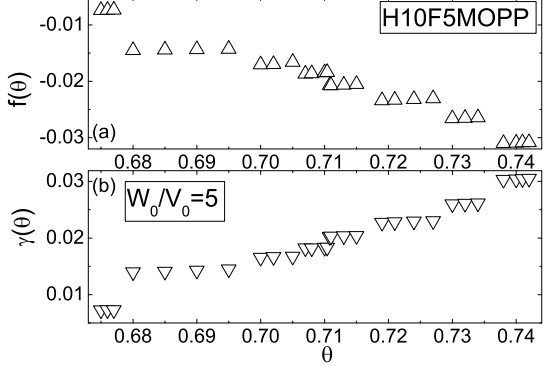

**Figure 15.** Plot of $f(\theta)$ (**a**) and $\gamma(\theta)$ (**b**) per H10F5MOPP molecule vs. $\theta$ [11], corresponding to a sequence of layer-thinning $25 \to 13 \to 11 \to 10 \to 9 \to 8 \to 7 \to 6$.

Calculations showed that above $\theta \sim 0.675$ ($\sim 358$ K), both the 25-layer free energy $f(\theta)$ and surface tension $\gamma(\theta)$ profiles demonstrate monotonic growth of these values with the smaller slope and abruptly jumps to the lower and higher values, respectively, at $\theta_{AI}(N = 25) \sim 0.678$ ($\sim 359.6$ K), where the film thins to 13 layers. This effect tends to repeat for the rest sequence of layer-thinning transitions $13 \to 11 \to 10 \to 9 \to 8 \to 7 \to 6$, where each thinning is characterized by abrupt jumps to the lower or higher values, both for $f(\theta)$ and $\gamma(\theta)$, respectively, and then continues to increase with the smaller, practically constant, positive slope. Please note that the value of $\gamma(25)$ per H10F5MOPP molecule of the 25-layer film is lower by a factor $\sim 4$ than the value of $\gamma(6)$ per molecule of the 6-layer film, where the numbers correspond to the successive layer-thinning process described earlier. In order to carry out the direct comparison between the high-resolution measured data on the surface tension $\Gamma$ [9,10] and calculated, in the framework of the extended McMillan's approach "enhanced" by anisotropic interactions in the bounding layers, with $W_0 = 5 V_0$, value on $\gamma(\theta)$ [11], the dimensionless value of $\gamma$ has been obtained. The calculated value of $\gamma(\theta = 0.67)$, for the case of H10F5MOPP 25-layer at temperature $\theta = 0.67$ ($\sim 353$ K), is equal to 0.0073 or $\Gamma = 0.0181$ N/m, while the measured value of $\Gamma$ is equal to 0.014 N/m. Hence, it has been obtained a good agreement between the theoretically predicted [11] and experimentally obtained [9,10] results.

It should be noted that the calculated values of surface tension show relatively small decrease of $\Gamma(N)$ as the film thins in correspondence to the sequence of the layer-thinning transitions $25 \to 13 \to 11 \to 10 \to 9 \to 8 \to 7 \to 6$. Calculations also showed [11] that the corresponding average values of $\Gamma(N)$ are: $\Gamma(25) \sim 0.0181$, $\Gamma(13) \sim 0.018$, $\Gamma(11) \sim 0.0179$, $\Gamma(10) \sim 0.0179$, $\Gamma(9) \sim 0.0178$, $\Gamma(8) \sim 0.0175$, $\Gamma(7) \sim 0.0174$, and $\Gamma(6) \sim 0.0172$. Here all data are in [N/m] [12].

These results show that the extended McMillan's approach "enhanced" by anisotropic interactions in the bounding layers is suitable for describing both the structural and thermodynamic properties of a partially fluorinated H10F5MOPP smectic film through the sequence of the abovementioned layer-thinning transitions.

"To examine the external field $\Delta$'s effect on the layer-thinning transition sequence and both on $f(\theta)$ and $s(\theta)$, calculations of the above values, for the case when the electric field **E** is directed across the film [11,47], has been carried out. Calculations showed that the dimensionless field $\Delta$'s effect on the layer-thinning sequence is reflected in the change of the layer-thinning transition sequences and of both values of the first multilayer jumps in the thickness and the corresponding layer-thinning temperatures $\theta_{AI}(N)$. For instance [11,47], in the case of $\Delta = 0.02$ the corresponding layer-thinning temperatures $\theta_{AI}(N)$ are: $\theta_{AI}(N = 25) \sim 0.7$ ($\sim 371.3$ K), $\theta_{AI}(N = 12) \sim 0.702$ ($\sim 372.4$ K), $\theta_{AI}(N = 11) \sim 0.704$ ($\sim 373.4$ K), $\theta_{AI}(N = 10) \sim 0.7075$ ($\sim 375.3$ K), $\theta_{AI}(N = 9) \sim 0.711$ ($\sim 377.1$ K), $\theta_{AI}(N = 8) \sim 0.719$ ($\sim 381.4$ K), $\theta_{AI}(N = 7) \sim 0.727$ ($\sim 385.6$ K), and $\theta_{AI}(N = 6) \sim 0.76$ ($\sim 403.1$ K), whereas in the case of $\Delta = 0.08$ the corresponding layer-thinning temperatures $\theta_{AI}(N)$ are: $\theta_{AI}(N = 25) \sim 0.704$ ($\sim 373.4$ K), $\theta_{AI}(N = 14) \sim 0.707$ ($\sim 375$ K), $\theta_{AI}(N = 12) \sim 0.709$ ($\sim 376.1$ K), $\theta_{AI}(N = 10) \sim 0.713$ ($\sim 378.2$ K), $\theta_{AI}(N = 9) \sim 0.716$ ($\sim 379.8$ K), $\theta_{AI}(N = 8) \sim 0.723$ ($\sim 383.5$ K), $\theta_{AI}(N = 7) \sim 0.731$ ($\sim 383.5$ K), and $\theta_{AI}(N = 6) \sim 0.76$ ($\sim 403.1$ K), respectively. In these calculations the following model parameters were chosen as $\alpha = 1.05$ and $W_0/V_0 = 5$. Therefore, in the case of $\Delta = 0.0$, the first thinning transition from $25 \to 13$ takes place at $\theta_{AI}(N = 25) \sim 0.678$ ($\sim 359.6$ K), whereas with the growth of $\Delta$ up to 0.08, the first thinning transition from $25 \to 14$ takes place at $\theta_{AI}(N = 25) \sim 0.704$ ($\sim 373.4$ K), what is on 13.8 K higher than in the case of $\Delta = 0.0$. Both the temperature $\theta'$ and field $\Delta$'s effects on the dimensionless Helmholtz free energy $f(\theta)$ in the smectic-A film, for the cases of $\Delta = 0.02$, 0.04, and 0.08, is shown in Figure 16a–c, respectively.

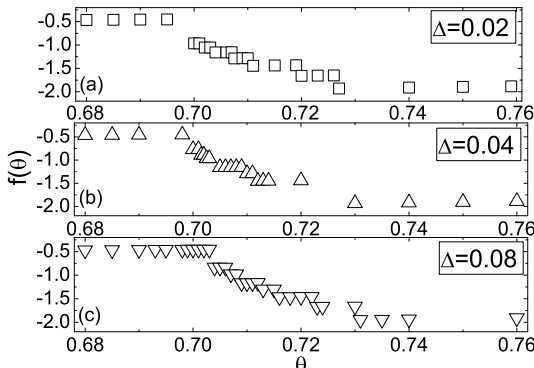

**Figure 16.** Plot of $f(\theta)$ per H10F5MOPP molecule vs. $\theta$ [11], for the cases $\Delta = 0.02$ (**a**), 0.04 (**b**), and 0.08 (**c**), corresponding to the sequences of layer-thinning transitions $25 \rightarrow 12 \rightarrow 11 \rightarrow 10 \rightarrow 9 \rightarrow 8 \rightarrow 7 \rightarrow 6$, $25 \rightarrow 15 \rightarrow 13 \rightarrow 12 \rightarrow 10 \rightarrow 9 \rightarrow 8 \rightarrow 6$, and $25 \rightarrow 14 \rightarrow 12 \rightarrow 10 \rightarrow 9 \rightarrow 8 \rightarrow 7 \rightarrow 6$, respectively.

Above $\theta \sim 0.68$ ($\sim 359.6$ K) the 25-layer free energy $f(\theta)$ per H10F5MOPP molecule increases with the smaller slope and abrupt jumps to the lower values at $\theta_{\mathrm{AI}}(\mathrm{N} = 25, \Delta = 0.02) \sim 0.7$ ($\sim 371.3$ K), $\theta_{\mathrm{AI}}(\mathrm{N} = 25, \Delta = 0.04) \sim 0.7$ ($\sim 371.3$ K), and $\theta_{\mathrm{AI}}(\mathrm{N} = 25, \Delta = 0.08) \sim 0.704$ ($\sim 373.4$ K), where the film thins to 12, 15, and 14 layers, respectively. This effect tends to repeat for the rest sequence of layer-thinning transitions $12 \rightarrow 11 \rightarrow 10 \rightarrow 9 \rightarrow 8 \rightarrow 7 \rightarrow 6$, for $\Delta = 0.02$, $15 \rightarrow 13 \rightarrow 12 \rightarrow 10 \rightarrow 9 \rightarrow 8 \rightarrow 7 \rightarrow 6$, for $\Delta = 0.04$, and $14 \rightarrow 12 \rightarrow 10 \rightarrow 9 \rightarrow 8 \rightarrow 7 \rightarrow 6$, for $\Delta = 0.08$, respectively, where each thinning is characterized by abrupt jump to the lower values of $f(\theta)$, and then continues to increase with the smaller, positive slope. Both the temperature $\theta'$ and field $\Delta$'s effects on the $\gamma(\theta)$ per H10F5MOPP molecule in the smectic-A film, for the cases of $\Delta = 0.02$, 0.04, and 0.08, is shown in Figure 17a–c, respectively.

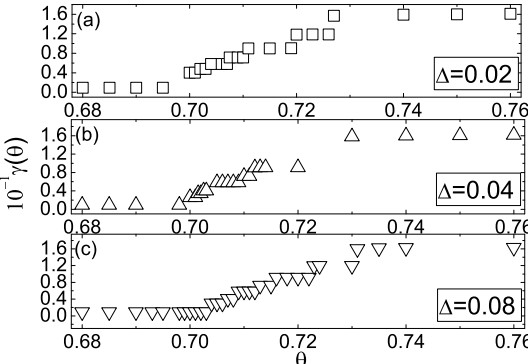

**Figure 17.** Plot of $\gamma(\theta)$ per H10F5MOPP molecule vs. $\theta$ [11], for the cases of $\Delta = 0.02$ (**a**), 0.04 (**b**), and 0.08 (**c**), corresponding to the same as in Figure 2 sequences of layer-thinning transitions.

These calculations showed that at each thinning the film tension $\gamma(\theta)$ abruptly jumps to the higher value and then continues to increase with the smaller slope [11,47], with growth of $\theta$ within the temperature interval $\theta_{\mathrm{AI}}(\mathrm{N}) < \theta < \theta_{\mathrm{AI}}(\mathrm{N} - \mathrm{n})$. The calculated data both on the dimensionless $\gamma(\Delta)$ and dimension $\Gamma(\Delta)$ surface tension per H10F5MOPP molecule for the 25-layer film [11], vs. $\Delta$, at the fixed temperature 359 $K$, are collected in Table 2 [11].

**Table 2.** The calculated $\gamma$ and measured $\Gamma$ values of the surface tension in free-standing partially fluorinated H10F5MOPP 25-layer smectic film vs. $\Delta$ [11].

| $\Delta$ | 0.00 | 0.02 | 0.04 | 0.08 |
|---|---|---|---|---|
| $\gamma$ | 0.0081 | 0.00922 | 0.0093 | 0.00937 |
| $\gamma$ [$N/m$] | 0.0178 | 0.0224 | 0.0226 | 0.0228 |

According to these calculations, the $\Delta$'s effect is characterized by increase of $\gamma(\Delta)$ up to 29% with increasing of $\Delta$ from 0.0 up to 0.08.

We can now estimate the magnitude of the electric field **E** necessary for the experimental observation of the effect of $\Delta$ on the change of the first layer-thinning transition temperature from $\theta_{AI}(N = 25) \sim 0.678$ ($\sim 359.6$ K), in the case of $\Delta = 0.0$, to $\theta_{AI}(N = 25) \sim 0.7$ ($\sim 371.3$ K), in the case of $\Delta = 0.02$. It can be obtained by applying the electric field $E \sim 1.36 \times 10^{-2}$ [C/m$^2$] across the 25-layer smectic-A film.

Therefore, based on these calculations one may conclude that the external electric field may affect not only the layer-thinning transition sequences, but also the change of the first multilayer jump in the film thickness and increase the value of the surface tension. Later effect is caused by enhancing of the order in the surface layers under the influence of the electric field applied across the layers.

These results indicate that the mean-field approach based on the extended McMillan's theory can be usefully applied for describing the effect of the external electric field both on the layer-thinning transitions and surface tension of free-standing smectic films. It has been shown, by solving the self-consistent nonlinear equations for the order parameters, that for the regime of strong interaction with $W_0/V_0 = 5$, both the layer-thinning transition temperatures and values of the surface tension grow with increasing of $\Delta$. Taking into account that there is good agreement between theoretical predictions and experimental results, this work lends credibility to the theoretical interpretation of the surface tension data and to the validity of the mean-field approach."[11].

The number of the optical techniques, such as the measurements of the optical transmission spectra [6] or the optical reflectivity [1,2,6] of thin smectic films, are the most effective experimental techniques which allow us to study these smectic films with the high-resolution. Measurements of the optical reflectivity $\mathcal{R}$ of thin smectic film composed of partially fluorinated molecules (H10F5MOPP) revealed a remarkable phenomenon of layer-thinning melting in smectic films upon heating above $\theta_{AI}(\text{bulk})$ [1]. It was shown that the experimentally obtained values of $\mathcal{R}(\theta)$ decrease in a series of sharp steps separated by plateaus as the temperature is increased [1,6]. In turn, the theoretical description of reflectivity $\mathcal{R}(\theta)$, which has been done in the framework of mean-field approach [8,47], shows that the values of $\mathcal{R}(\theta)$ also decrease in a series of the stepwise reduction of reflectivity when the temperature is increased above $\theta_{AI}(\text{bulk})$. In the limiting case, when the smectic film is sufficiently thin and the wavelength $k_0$ of incident radiation is within the visible range, the reflectivity $\mathcal{R}$ can be written as [8,47]

$$\mathcal{R} = \frac{k_0^2}{4}\left[\sum_{i=1}^{N}\left(n_i^2 - 1\right)\mathcal{L}_i\right]^2, \tag{16}$$

where the refractive indices $n_i^2$ can be expressed in terms of the OP $\eta_i$ and the film thickness $\mathcal{L}_i$, corresponding to the $i$th layer. In turn, the film thickness $\mathcal{L}_i$ can be found as [7,8]

$$\mathcal{L}_i(N, T) = \mathcal{L}_0\left[1 - \frac{\mathcal{P}(N, T)}{B_i(N, T)}\right], \tag{17}$$

where

$$\mathcal{P}(N, T) = -\frac{\Delta F(N, T)}{\mathcal{L}(N)} \tag{18}$$

is the disjoining pressure acting on the film layers from the bounding surfaces, and

$$B_i(N, T) = B_0\left[\frac{\sigma_i(N, T)}{\sigma_b(T)}\right]^2 \tag{19}$$

is the compressibility modulus of the $i$th layer. Here $\mathcal{L}(N) = \frac{1}{N}\sum_{i=1}^{N}\mathcal{L}_i$, where $\mathcal{L}_i$ is the thickness of the $i$th layer, $\sigma_i(N, T)$ and $\sigma_b(T)$ are the values of the translational OPs corresponding to the $i$th layer

and the bulk SmA phase, respectively, and $B_0$ and $\mathcal{L}_0$ are the compressibility modulus and the layer thickness in the absence of the disjoining pressure, respectively. It should be pointed out that the set of translational OP $\sigma_i$, corresponding to the $i$th layer, as well as the change $\Delta F = F(N, T) - F((N-1), T)$ of the total Helmholtz free energy of the smectic film, can be calculated in the framework of the abovementioned mean-field theory [7,8,12]. Please note that the change of the total Helmholtz free energy $\Delta F$ of the smectic film is equal to the work which must be performed on the film unit surface area to decrease its thickness by one layer. Here $F(N, T)$, is the full Helmholtz free energy corresponding to the $N$-layer smectic film. In principle, two variants can be realized, the first variant is when the value of $\Delta F$ is positive, then the disjoining pressure $\mathcal{P}$ prevents the thinning of FSSF, and the film layers are subjected to a stretching force. On the other hand, when the value of $\Delta F$ is negative, the disjoining pressure promotes a thinning of the smectic film, and its layers are subjected to a compressive force. Calculations showed that both the $\eta_i$ and $\sigma_i$ OPs for smectic layers demonstrate strong ordering in the bounding domains, and the profiles of $\eta_i$ and $\sigma_i$ are characterized by rapid decrease of both OPs with distance from those surfaces. This nonuniformity of the film was taken into account when the reflectivity and layer- thinning compression have been computed.

The electric field $\Delta$'s effect on the smectic layers should give rise to the change of their thicknesses $\mathcal{L}_i(N, \theta, \Delta)$. According to Equations (17)–(19), the thickness $\mathcal{L}_i(N, \theta, \Delta)$ of the $i$th film layer is the function of the disjoining pressure $\mathcal{P}(N, \theta, \Delta)$ and the compressibility modulus $B_i = B_0 \left[\sigma_i / \sigma(\text{bulk})\right]^2$ for each layer $i$ of FSSF of a given thickness $N$.

The temperature $\theta'$s effect on the dimensionless disjoining pressure $P(\theta, \Delta) = \mathcal{P}(\theta, \Delta)/V_0 n_0$, investigated in the framework of the mean-field approach, is shown in Figure 18.

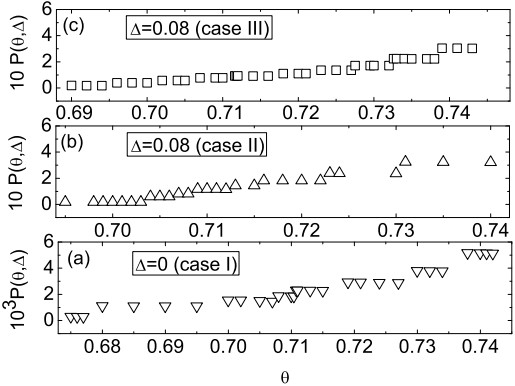

**Figure 18.** Plot of $P(\theta, \Delta)$ vs. $\theta$ [47] during the layer-thinning transitions corresponding to cases I (**a**), II (**b**), and III (**c**), respectively.

These calculations correspond to the sequences of the layer-thinning transitions, for three cases [47]: $\Delta = 0$ (case I), $\Delta = 0.08$ $\left(\mathbf{E} \parallel \hat{\mathbf{k}}\right)$ (case II), and $\Delta = 0.08$ $\left(\mathbf{E} \parallel \hat{\mathbf{i}}\right)$ (case III), respectively, and showed that the external electric field ($\Delta = 0.08$), both directed across (case II) and along (case III) smectic film, has the strong influence on $P(\theta, \Delta)$. Indeed, in both cases II and III, the values of $P(\theta, \Delta = 0.08)$ (Figure 5b,c) are on average by two orders of magnitude greater than the value of $P(\theta, \Delta = 0)$ (Figure 18a) for the case I. Please note that the nature of such the electric field $\Delta$'s effect on $P(\Delta)$ is due to the $\Delta$'s effect on the Helmholtz free energy $f(\Delta)$. Indeed, the values of $f(\Delta = 0.08)$ are approximately by one order of magnitude greater than the values of $f(\Delta = 0)$ (see Figures 15 and 16), when the electric field is absent. As a result, anyone can find that the values of $P(\Delta = 0.08)$ are on

average by two orders of magnitude greater than the value of $P(\Delta = 0)$. This means that the average dimensional disjoining pressure $\mathcal{P}$ in the smectic film with $N = 25$ layers is [47]

$$
\mathcal{P} \sim
\begin{cases}
6.6 \times 10^5 \ N/m^2, & \text{for } \Delta = 0.08, \ \mathbf{E} \parallel \hat{\mathbf{k}}, \\
4.3 \times 10^5 \ N/m^2, & \text{for } \Delta = 0.08, \ \mathbf{E} \parallel \hat{\mathbf{i}}, \\
3.1 \times 10^3 \ N/m^2, & \text{for } \Delta = 0.
\end{cases}
$$

Based on these calculations, one can conclude that the layer-thinning transitions are characterized by abrupt (stepwise) increase of $\mathcal{P}(N)$ when the film thins from $N$-layer to $(N-1)$-layer film, then from $(N-1)$-layer to $(N-2)$-layer film, and so on. All smectic layers during the thinning process are subjected to the compressive force which grows with $N$ as [12] $\mathcal{P}(N) \sim \mathcal{O}\left(\frac{1}{N}\right)$. The electric field $\Delta$'s effect on the smectic layers should give rise to the change of their dimensionless thicknesses $L_i(N, \theta, \Delta) = \mathcal{L}_i(N, \theta, \Delta) / \mathcal{L}_0$. The behavior of the dimensionless smectic layer thickness profiles $L_i(N = 25, \Delta)$ across the 25-layer partially fluorinated H10F5MOPP smectic film, for several values of $\Delta$ [47], showed that the interior film layers are compressed much stronger than the bounding layers. In the case of $\Delta = 0.0$, the interior layers are compressed weaker than in cases when the electric field is applied. Calculations also showed that with decreasing of the film thickness the biggest compressions of interior layers are increased, from $L_{i=13}(N = 25) \sim 0.98$, for 25-layer film, to $L_{i=5}(N = 10) \sim 0.845$, for 10-layer film, respectively. Physically, this means that in the case of thinner films all the layers are subjected to bigger compressive forces than in the case of thicker ones. The dimensionless field $\Delta$'s effect on the average film thicknesses $L(\theta, \Delta)$ in the smectic film corresponding to the sequence of the abovementioned layer-thinning transitions is shown in Figure 19, and characterized by the stepwise decreasing of $L(\theta, \Delta)$ [8,47].

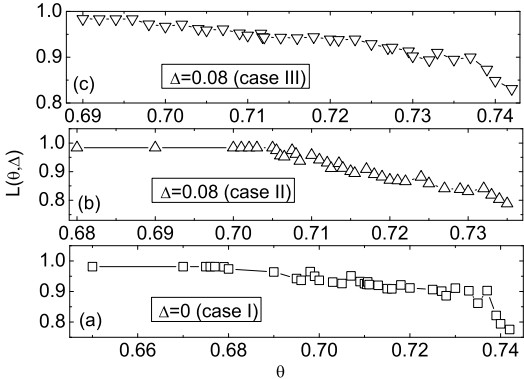

**Figure 19.** Plot of $L(\theta, \Delta)$ vs. $\theta$, for the free-standing partially fluorinated H10F5MOPP smectic film at each thinning, for cases I ($\Delta = 0$) (**a**), II ($\Delta = 0.08$) (**b**), and III ($\Delta = 0.08$) (**c**), respectively.

Calculations showed that at each thinning the film thickness abruptly jumps to the higher values and then continues to decrease with the smaller slope, with growth of $\theta$ within the temperature interval $\theta_{AI}(N) < \theta < \theta_{AI}(N - n)$. Here $N - n$ is the number of smectic layers remaining in the film after each thinning.

Behavior of the average film thicknesses $\mathcal{L}(T)$ measured in the smectic film composed of 2-(4-(1,1-dihydro-2-(2-perfluorobutoxy) perfluoroethoxy) perfluoroethoxy) phenyl-5-octyl pyrimidine (H8F(4,2,1)MOPP) molecules also exhibit the upward jumps at each thinning transition [33]. These results show that the extended McMillan's approach "enhanced" by anisotropic interactions in the bounding layers is suitable for describing the stepwise reductions of the smectic film thickness through the sequence of the abovementioned layer-thinning transitions. Hence, it has been obtained a good agreement between the theoretically predicted [8] and experimentally observed decrease of $L(\theta)$ with decrease of $N$, for the FSSF composed of partially fluorinated molecules H8F(4,2,1)MOPP.

The understanding of how the temperature $\theta$ and the electric field $\Delta$ effects on the reflectivity $\mathcal{R}(\theta, \Delta)$ in the smectic film [47] through the sequence of the abovementioned layer-thinning transitions, has been obtained in the framework of the mean-field approach [47]. The calculation results are shown in Figure 20 and indicate that the reflectivity also demonstrates the stepwise reductions of $\mathcal{R}(\theta, \Delta)$ during the sequence of the abovementioned layer-thinning transitions. Plot of $R(\theta, \Delta) = \mathcal{R}(\theta, \Delta) / \mathcal{L}_0^2 k_0^2$ vs. $\theta$, for the case II, and several values of $\Delta$ [47] is shown in Figure 20. Here the set of $\Delta$ values are: 0 (a), 0.02 (b), 0.04 (c), and 0.08 (d), respectively.

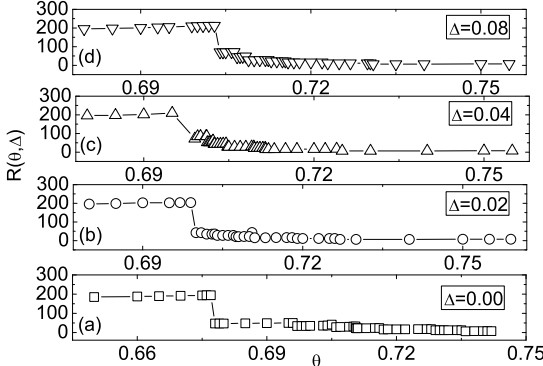

**Figure 20.** Plot of $R(\theta, \Delta)$ vs. $\theta$, for the case II, and for several values of $\Delta$ [47]: 0.00 (**a**), 0.02 (**b**), 0.04 (**c**), and 0.08 (**d**), respectively.

These results indicate that the mean-field approach based on the extended McMillan's theory can be usefully applied for describing not only the layer-thinning transitions which occurs through the series of layer-thinning causing the films to thin in a stepwise manner as the temperature is increased above $\theta_{AI}$(bulk), but also several structural, thermodynamic and optical properties of free-standing smectic films. Taking into account that there is a good agreement between theoretical predictions and experimental results, this mean-field approach lends credibility to the theoretical interpretation of a wide range of structural and optical data.

In the next Section the diffusion phenomena in thin smectic films will be discussed.

### 2.2.6. Translational and Orientational Diffusion across the Smectic Films

Although several approaches have been proposed to theoretically describe the diffusion process in liquid crystals [48–51], it is still too early to talk about the development of a theory which would make it possible to describe the diffusion processes in thin smectic films based only on the form of the Hamiltonian. In the bulk of the SmA phase the translational diffusion process across the smectic layers implies a passage through a potential barrier $\Phi$. Taking into account that in the smectic-A phase the coordinate system is chosen so that the direction of $z$-axis coincides with direction of the director $\hat{\mathbf{n}}$, the potential barrier $\Phi(z + d) = \Phi(z)$ is a periodic function of $z$, with the period $d$, which is the layer spacing. The jump rate for molecular diffusion in the bulk of the SmA phase can be described, for instance, by the translational diffusion model [52], which assumes a stochastic Brownian process, in which each molecule moves in time as a sequence of small steps caused by collisions with its surrounding molecules and under the influence of the potential $\Phi(z)$, which is set up by these molecules. This diffusional process can be described by the translational diffusion tensor whose principal elements ($\mathrm{D}_{xx} = \mathrm{D}_{yy} = \mathcal{D}_\perp$, $\mathrm{D}_{zz} = \mathcal{D}_\parallel$) are determined in a frame fixed on the molecule.

Recently, a molecular model based upon the random walk theory [52] has been proposed to describe translational diffusion in freely suspended smectic films [53]. It was shown that for the calculation of the translational diffusion coefficient (TDC) $\mathcal{D}_\parallel$ across the smectic layers both in the bulk

of the film, as well as in the vicinity of the bounding surfaces the set of $\eta_i$ and $\sigma_i$ OPs, obtained by using the mean-field McMillan's approach [7] with anisotropic forces [39] are required.

"The random walk theory allows us to calculate the translational diffusion across the smectic layers when a molecule makes a jump from $(i+1)$th to $i$th layer. It can be realized when the molecule reaches "the boundary" between these layers with a "positive" momentum. Here the layers are counted from the film/air interface to the bulk of the film. In that case, the TDC can be written as [53]

$$\mathcal{D}_{i,i+1} = \mathcal{D}_{\parallel}^{i,i+1} = \frac{\langle z_{i,i+1}^2 \rangle}{\tau_{i,i+1}},\tag{20}$$

where $\langle z_{i,i+1}^2 \rangle = \langle z^2 \rangle = \int_0^d z^2 h(z)\, dz$ is the mean-square jump length from $(i+1)$th to $i$th layers, $h(z)$ is the one-particle distribution function (see Equation (5)), and $\tau_{i,i+1}$ is the time required the molecule to jump from $(i+1)$th layer to the $i$th ones. In turn, the time $\tau_{i,i+1}$ can be written as

$$\tau_{i,i+1} = \tau_0 \exp\left[\frac{\Delta\Phi_{i,i+1}}{k_B T}\right],\tag{21}$$

where $\tau_0$ is the time of oscillation of the molecule about the equilibrium position in the bulk of the smectic film, and

$$\Delta\Phi_{i,i+1} = \Phi_i(\max) - \Phi_{i+1}(\min)\tag{22}$$

is the height of the potential barrier. Here $\Phi_i(\max)$ and $\Phi_{i+1}(\min)$ are the values of maxima and minima neighborhood potentials belonging to $i$th and $(i+1)$th layers, respectively. For calculation of the potential barrier $\Delta\Phi_{i,i+1}$ one needs an effective anisotropic periodic potential

$$\Phi_i(z_i) = \int_{-1}^{+1} d(\cos\theta_i)\, \Phi_i(z_i,\theta_i)\tag{23}$$

within the $i$th smectic layer. By implementing the integration in the last equation one obtains the set of expressions for the height of the potential barrier

$$-\frac{3}{V_0}\Phi_1(z_1) = \frac{W_0}{V_0}\eta_1 + \eta_2 + \Delta\left(\eta_1 + \frac{1}{2}\right) + \alpha\cos(2\pi z_1)\left(\frac{W_0}{V_0}\sigma_1 + \sigma_2\right),$$

$$-\frac{3}{V_0}\Phi_{1<i<N}(z_i) = \sum_{j=i-1}^{i+1} q_j + \Delta\left(\eta_i + \frac{1}{2}\right) + \alpha\cos(2\pi z_i)\sum_{j=i-1}^{i+1}\sigma_j,$$

$$-\frac{3}{V_0}\Phi_N(z_N) = \frac{W_0}{V_0}\eta_N + \eta_{N-1} + \Delta\left(\eta_N + \frac{1}{2}\right) + \alpha\cos(2\pi z_N)\left(\frac{W_0}{V_0}\sigma_N + \sigma_{N-1}\right),\tag{24}$$

where $\bar{z}_i = z_i/d$ is the dimensionless space variable. Notice that the overbar in the space variable $z$ has been (and will be) eliminated in the last as in the following equations.

Furthermore, it is convenient to rewrite expressions for potential barriers as [53]

$$-\frac{3\Delta\Phi_{1,2}}{V_0} = -\frac{3}{V_0}\left(\Phi_1\left(\max\right) - \Phi_2\left(\min\right)\right) =$$

$$= \frac{W_0}{V_0}\eta_1 + \eta_2 + \alpha\left(\frac{W_0}{V_0}\sigma_1 + \sigma_2\right) + \Delta\left(\eta_1 - \eta_2\right)$$

$$-\eta_1 - \eta_2 - \eta_3 + \alpha\left(\sigma_1 + \sigma_2 + \sigma_3\right),$$

$$-\frac{3\Delta\Phi_{i,i+1}}{V_0} = -\frac{3}{V_0}\left(\Phi_i\left(\max\right) - \Phi_{i+1}\left(\min\right)\right) =$$

$$= \eta_{i+1} + \eta_i + \eta_{i-1} + \alpha\left(\sigma_{i+1} + \sigma_i + \sigma_{i-1}\right) + \Delta\left(\eta_i - \eta_{i+1}\right)$$

$$-\eta_i - \eta_{i+1} - \eta_{i+2} + \alpha\left(\sigma_i + \sigma_{i+1} + \sigma_{i+2}\right),$$

$$-\frac{3\Delta\Phi_{N,N-1}}{V_0} = \frac{3}{V_0}\left(\Phi_N\left(\max\right) - \Phi_{N-1}\left(\min\right)\right) =$$

$$= \frac{W_0}{V_0}\eta_N + \eta_{N-1} + \alpha\left(\frac{W_0}{V_0}\sigma_N + \sigma_{N-1}\right) + \Delta\left(\eta_{N-1} - \eta_N\right)$$

$$-\eta_N - \eta_{N-1} - \eta_{N-2} + \alpha\left(\sigma_N + \sigma_{N-1} + \sigma_{N-2}\right). \tag{25}$$

Having obtained the set of OPs $\eta_i$ and $\sigma_i$ $(i = 1, ..., N)$ one can calculate the potential barrier $\Delta\Phi_{i,i+1}$, the mean-square jump length from $(i + 1)$th to $i$th layers [53]

$$\langle z_{i,i+1}^2 \rangle = \langle z^2 \rangle = \int_0^1 z^2 h\left(z\right) dz, \tag{26}$$

and the TDC $\mathcal{D}_{i,i+1} = \mathcal{D}_{\parallel}^{i,i+1}$.

The calculated values of $D_{i,i+1}\left(\theta, \Delta = 0\right) / D_{N/2,N/2+1}\left(\theta, \Delta = 0\right)$ vs. the number $i$, in the smectic film with $N = 25$ layers and in the absence of the electric field ($\mathbf{E} = 0$) are shown in Figure 21.

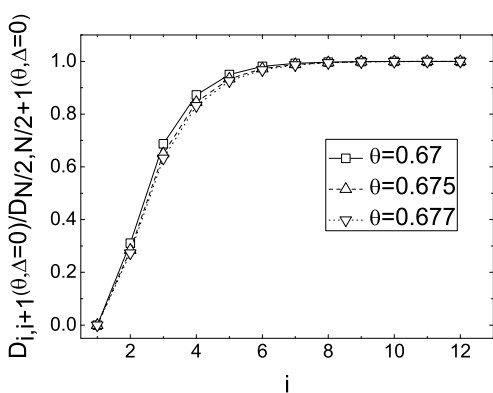

**Figure 21.** Plot of $D_{i,i+1}\left(\theta, \Delta = 0\right) / D_{N/2,N/2+1}\left(\theta, \Delta = 0\right)$ vs. $i$ [53] in the absence of the electric field ($\mathbf{E} = 0$), and for three values of $\theta$: 0.67 (squares), 0.675 (up triangles), and 0.677 (down triangles), respectively.

Calculations showed [53] that the distribution of the profiles $D_{i,i+1}\left(\theta, \Delta = 0\right) / D_{N/2,N/2+1}\left(\theta, \Delta = 0\right)$ across the 25 layer smectic film, in the absence of the electric field ($\Delta = 0.0$), corresponding to three temperature $\theta$ values 0.67 (squares), 0.675 (up triangles), and 0.677 (down triangles), respectively, are characterized by the monotonic increase of the ratio $D_{i,i+1}\left(\theta, \Delta = 0\right) / D_{N/2,N/2+1}\left(\theta, \Delta = 0\right)$ up to the middle-film's values, with increasing distance (or number of layers) from the bounding surface towards the interior of the film. In the case of strong ($W_0 = 5V_0$) "enhanced" pair interactions in the bounding layers these distributions demonstrate monotonic growth of the value of $D_{i,i+1}\left(\theta, \Delta = 0\right) / D_{N/2,N/2+1}\left(\theta, \Delta = 0\right)$ up to the eighth layer from each boundary, where the function $D_{i,i+1}\left(\theta, \Delta = 0\right) / D_{N/2,N/2+1}\left(\theta, \Delta = 0\right)$ saturates and does

not change with further increase of $i$. In turn, near the bounding surface the motional constant $D_{i,i+1}(\theta, \Delta = 0) / D_{N/2,N/2+1}(\theta, \Delta = 0)$ drops to zero, i.e., the strong "enhanced" pair interactions completely suppresses the diffusion process in the bounding layers. The distribution of the number of $D_{i,i+1}(\theta, \Delta = 0) / D_{N/2,N/2+1}(\theta, \Delta = 0)$ profiles across the smectic films, during the sequence of the layer-thinning transitions $25 \to 13 \to 11 \to 10$ [53], as the temperature is increased above the value $\theta_{AI}(\text{bulk}) \sim 0.675$, is shown in Figure 22.

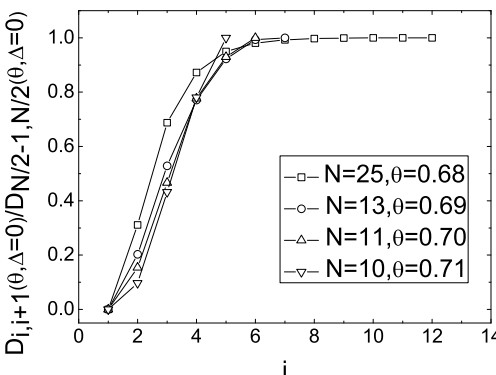

**Figure 22.** Plot of $D_{i,i+1}(\theta, \Delta = 0) / D_{N/2,N/2+1}(\theta, \Delta = 0)$ vs. $i$, in the absence of the electric field ($\Delta = 0.0$). Here the sequence of the layer-thinning transitions is $25 \to 13 \to 11 \to 10$ [53].

Here, calculations have been carried out in the absence of the electric field ($\Delta = 0.0$) [53]. The electric field **E**'s effect on the dimensionless translational diffusion coefficient $D_{i,i+1}(\theta, \Delta) / D_{N/2,N/2+1}(\theta, \Delta)$ as a function of layer number $i$, in the smectic film with $N = 25$ layers, both in the cases of $\Delta = 0.0$ and $\Delta = 0.08$) [53], is shown in Figure 23.

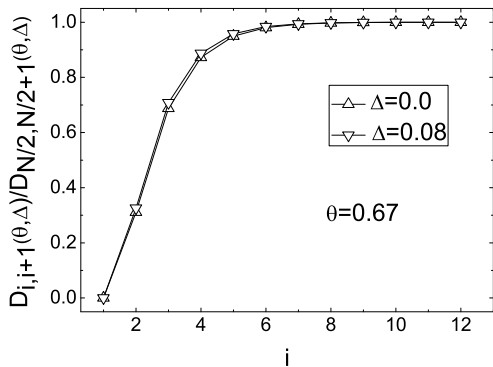

**Figure 23.** Plot of $D_{i,i+1}(\theta, \Delta) / D_{N/2,N/2+1}(\theta, \Delta)$ vs. $i$, for two values of $\Delta$: first, for $\Delta = 0.0$, and second, for $\Delta = 0.08$ [53], respectively.

These calculations showed that the electric field $\Delta$ has a weak effect on distribution of $D_{i,i+1}(\theta, \Delta) / D_{N/2,N/2+1}(\theta, \Delta)$ across the 25-layer smectic film and the diffusion process is completely suppressed in the bounding layers [53]. Such behavior of the TDC is due to the fact that the potential barrier $\Delta\Phi_{1,2}$ is much bigger than $\Delta\Phi_{i,i+1}$ ($i = 2, ..., N/2$) as the temperature is increased above the bulk value $\theta_{AI}(\text{bulk})$, because of the strong ordering in the vicinity of the bounding layers.

It should be noted that the abovementioned mean-field model is applicable to describe diffusion across smectic layers, because one deals with the potential barrier which is set up across the smectic layers but not within the smectic layers.

To calculate the dimension value of translational diffusion coefficient $D_{\parallel}^b$ (N) in the smectic film one can use the Maclaurin expansion of momentum autocorrelation function $f(t) = \langle \vec{p}(0) \cdot \vec{p}(t) \rangle / \langle \vec{p}(0) \cdot \vec{p}(0) \rangle$. For such purposes, the function $f(t)$, in the form of damped oscillation, has been adopted for calculation of the dimensional value of diffusion coefficient [53]

$$D_{\parallel}^b (N) \equiv D_b (N) = \frac{k_B T}{m} \int_0^\infty f(t)\, dt, \tag{27}$$

where the angle brackets indicate the equilibrium ensemble average [52]. Taking into account that the coefficients of Maclaurin series in time of function $f(t)$ are in principle calculable [53], a two-parameter functional expression for $f(t)$ takes the form [54]

$$f(t) = \cos(\alpha \delta t) / \cosh(\alpha t), \tag{28}$$

where the parameter $\alpha$ determines the rate of decay, and $\delta$ gives the rate of oscillation relative to the time scale determined by $\alpha$. All this allows us to record the diffusion coefficient $D_b$ (N), in terms of $\alpha$ and $\delta$, which takes the form [54]

$$D_b (N) = \frac{k_B T}{m} \frac{\pi}{\alpha} \cosh^{-1} \left( \frac{\pi \delta}{2} \right), \tag{29}$$

where parameters $\alpha$ and $\delta$ are given in Ref. [53]. The calculated data on $D_b$ (N $= 25$), at $T = 367\,\text{K}$ ($\theta = 0.67$) and $d = 3.26$ nm [55], for 25-layer partially fluorinated H10F5MOPP smectic film, gives $\sim 6\,\mu\text{m}^2/\text{s}$. In turn, the experimentally obtained data on $D_b$ (N) ($N = 25, 13, 11, 10$), for 25-layer smectic film composed of 4-octyl-4′-cyanobiphenyl molecules, gives $\sim 3\,\mu\text{m}^2/\text{s}$ [56,57]. Hence, it has been obtained a good agreement between the theoretically predicted [53] and experimentally obtained [55] results." [53].

"In turn, the rotational dynamics of a uniaxial molecule in anisotropic phase can be described in the framework of the rotational diffusion model [58], which is based on the concept that the molecular reorientation proceeds through a random sequence of large-amplitude angular jumps from one orientation to another [51]. In that model, a molecule is considered to be an ellipsoid aligned along, or close to $\hat{\mathbf{n}}$, where the diffusional jump results in rotation of the molecule from $\beta \sim 0$ to $\beta \sim \pi$. This assumes that a molecule to make jump by a minimal successful angle $\pi$, if reaches "the boundary" between the $\beta_i \sim 0$ and $\beta_i \sim \pi$ orientations, and has a positive angular-momentum projection $p_\beta$ onto any axis perpendicular to $\hat{\mathbf{n}}$.. In the framework of this model, the rotational self-diffusion (RSD) coefficient $\mathcal{D}_\omega \equiv \mathcal{D}_\perp$ can be written as [51]

$$\mathcal{D}_\omega \equiv \mathcal{D}_\perp = \frac{1}{2} k_\omega \pi^2, \tag{30}$$

where

$$k_\omega = \frac{1}{\mathcal{I}} \int_{-\infty}^\infty dp_\varphi \int_0^\infty dp_\beta p_\beta \int_0^{2\pi} d\varphi \mathcal{F}(p_\varphi, p_\beta, \varphi, \beta)_{\beta = \frac{\pi}{2}} \tag{31}$$

is the rotational jump rate, $\mathcal{I}$ is the moment of inertia of the molecule with respect to the minor axis of the ellipsoid, $p_\beta$ and $p_\varphi$ are the angular-momentum components, $\mathcal{F}(p_\varphi, p_\beta, \varphi, \beta)$ is the one-particle distribution function of the LC film on the solid surface. The function $\mathcal{F}$ does not depend on the azimuthal angle $\varphi$, and, moreover, in the vicinity of the equilibrium state, the momentum projections are neither correlated between them nor with the conjugated angles $\varphi$ and $\beta$. In this case, the function $\mathcal{F}$ can be written as a product of three functions

$$\mathcal{F}(p_\varphi, p_\beta, \beta, \varphi) = \mathcal{F}(p_\varphi)\, \mathcal{F}(p_\beta)\, f(\cos \beta), \tag{32}$$

where $\mathcal{F}(p_\varphi)$ and $\mathcal{F}(p_\beta)$ are the Maxwellian distribution functions, whereas $f(\cos \beta)$ is the ODF. By integrating Equation (31) one obtains the final expression for the coefficient RSD [51,59,60]

$$\mathcal{D}_\omega = \pi^3 \sqrt{\frac{k_B T}{2\pi \mathcal{I}}} f\left(\frac{\pi}{2}\right) = \pi^3 \sqrt{\frac{\theta V_0}{6\pi \mathcal{I}}} f\left(\frac{\pi}{2}\right). \tag{33}$$

Thus, $\mathcal{D}_\omega$ is the function of temperature $\theta$, $\mathcal{I}$, and the value of the ODF at $\beta = \frac{\pi}{2}$. Physically, this means that the one-particle function $f(\cos \beta)$ of the LC phase has a rather sharp maximum at the point $\beta = 0$ (i.e., around the director $\hat{\mathbf{n}}$), rapidly decreasing as $\beta$ tends to $\frac{\pi}{2}$. At $\beta = \frac{\pi}{2}$ the function $f(\beta_i)$ is small but finite, and defines the "gate" width in orientational space through which the molecule diffuses from one orientation to another. Therefore, having obtained the ODF $f\left(\frac{\pi}{2}\right)$, one can calculate, using Equation (33), $\mathcal{D}_\omega$ as the function of $\theta$ and $\mathcal{I}$.

The numerical analysis of rotational diffusion processes in thin smectic film ($N = 25$) deposited on the solid surface (with $W_0/V_0 = 3.0$ and $W_1/V_0 = 10.0$) showed that only a strong electric field $\Delta = 0.1$ has a visible effect on the dimensionless coefficient [61] $\mathcal{D}_\omega(i, \theta)/\mathcal{D}_\omega$ (bulk) ($i = 1, 25$) (see Figure 24).

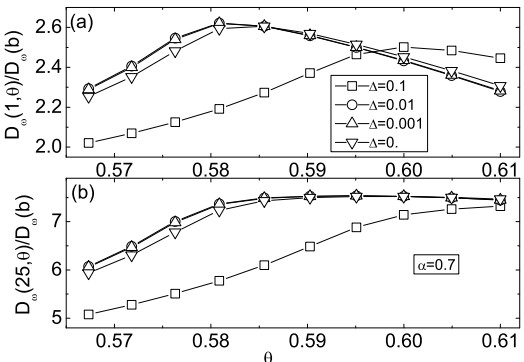

**Figure 24.** Plot of $\mathcal{D}_\omega(i, \theta)/\mathcal{D}_\omega$ (bulk) [61], (**a**) for $i = 1$, and (**b**) for $i = 25$, vs. $\theta$, for four values of $\Delta$: 0 (down triangles), 0.001 (up triangles), 0.01 (circles), and 0.1 (squares), respectively.

Here $W_0$ and $W_1$ are two parameters of the LC system which are defined for the enhanced pair interactions in the LC/vacuum and LC/solid bounding layers, respectively, and the dimensionless electric field $\Delta = 0.1$ can be obtained by applying the electric field $\sim 7 \times 10^{-2} \, \text{C/m}^2$ across the 25-layer smectic film. Calculations showed that the motional constant $\mathcal{D}_\omega(25, \theta)/\mathcal{D}_\omega$ (bulk) decreases in the low-temperature range ($0.57 \leq \theta \leq 0.59$), up to 20% [61], with increasing of $\Delta$ from 0 to 0.1, both in the first ($i = 1$, Figure 24a) and the last ($i = 25$, Figure 24b) layers [61]. In the high-temperature range ($0.59 \leq \theta \leq 0.61$) the effect of the electric field $\Delta$ decreases, and finally disappears at the end of the temperature interval ($0.57 \leq \theta \leq 0.61$). In the case of a strong electric field ($\Delta = 0.1$), calculations showed that curves describing $\mathcal{D}_\omega(i, \theta)/\mathcal{D}_\omega$ (bulk) vs. $\theta$, both for the first ($i = 1$) and the last ($i = 25$) layers, are practically congruent curves [61].

The parameter $\alpha$'s effect on $\mathcal{D}_\omega(i, \theta, \alpha)/\mathcal{D}_\omega$ (bulk) for two values of $\Delta$ is shown in Figures 25 and 26.

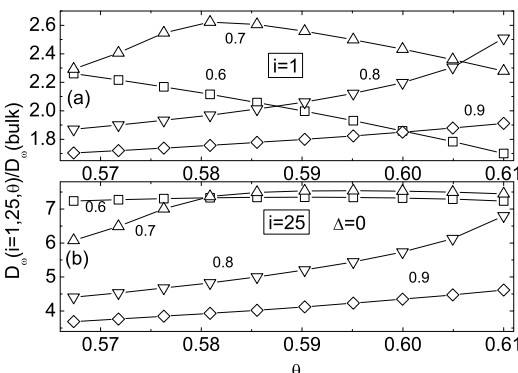

**Figure 25.** Plot of $\mathcal{D}_\omega (i, \theta) / \mathcal{D}_\omega$ (bulk) [61], (**a**) for $i = 1$, and (**b**) for $i = 25$, vs. $\theta$ for different values of parameter $\alpha$: 0.6 (squares), 0.7 (up triangles), 0.8 (down triangles), and 0.9 (rhombus), respectively.

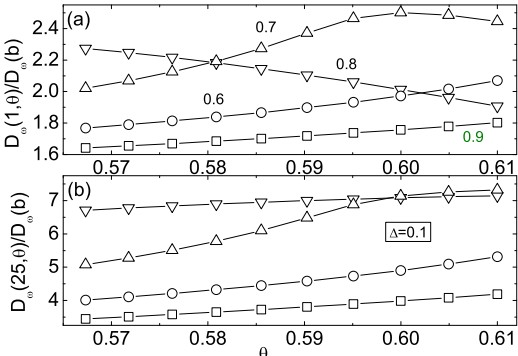

**Figure 26.** Same as in Figure 25, but the value of $\Delta$ is equal to 0.1.

Calculations showed [61] that in the case when the electric field is absent ($\Delta = 0$), and in the low-temperature limit $0.57 \leq \theta \leq 0.61$, the lower values of $\alpha$, i.e., 0.6 and 0.7, produce the higher values of $\mathcal{D}_\omega (25, \theta) / \mathcal{D}_\omega$ (bulk) (see Figure 26b), whereas in the first layer ($i = 1$), the lower values of $\alpha$ produce the higher dimensionless RSD coefficient only at the beginning of that temperature interval. At the end of the temperature interval $[0.57 \leq \theta \leq 0.61]$ the higher values of $\mathcal{D}_\omega (1, \theta) / \mathcal{D}_\omega$ (bulk) are produced at $\alpha = 0.8$ (see Figure 26a). In the case of a strong electric field ($\Delta = 0.1$), the dependence of $\mathcal{D}_\omega (i, \theta) / \mathcal{D}_\omega$ (bulk) vs. $\theta$ is shown in Figure 26, and demonstrates the same qualitative behavior as in the case when the electric field is absent, only in the last ($i = 25$) (see Figure 26b) layer. In the first layer ($i = 1$) and at the end of that temperature interval the higher values of $\mathcal{D}_\omega (1, \theta) / \mathcal{D}_\omega$ (bulk) are produced at $\alpha = 0.7$ (see Figure 26a). With the growth of the value of $\Delta$ from 0 to 0.1, the biggest values of $\mathcal{D}_\omega (i, \theta) / \mathcal{D}_\omega$ (bulk) ($i = 1, 25$) [61] are produced by the molecules with the lower values of the alkyl tail length $\alpha = 0.6, 0.7$, and 0.8. These calculations, based on the alkyl tail length $\alpha$'s effect on $\mathcal{D}_\omega (i, \theta, \alpha) / \mathcal{D}_\omega$ (bulk) ($i = 1, 25$), and displayed in Figures 24–26, showed that the parameter $\alpha$ has a strong effect on the rotational diffusion process in the smectic film deposited on the solid surface and subjected to the strong electric field. In all the cases described above, the value of the moment of inertia of the molecule with a change in $\alpha$, did not change [61]. Calculations of the coefficient $\mathcal{D}_\omega$ (bulk) in the bulk of the LC phase composed of 8CB molecules, at $T = 307$ K, with $I = 7.41 \times 10^{-44}$ [kg m$^2$] and $f \left( \frac{\pi}{2} \right) \sim 10^{-4}$, gives $\mathcal{D}_\omega$ (bulk) $\sim 3 \times 10^8$ s$^{-1}$, which is in a good agreement with experimental $^2$H NMR $(0.5 - 3) \times 10^8$ s$^{-1}$ [62] values. Please note that the function $f \left( \frac{\pi}{2} \right)$ has been obtained by solving the system of nonlinear Equations (3) and (4), for two bulk OPs $\eta_b$ and $\sigma_b$ with the effective anisotropic potential $\Phi (z, \cos \beta) = V_0 \left[ \eta_b + \alpha \sigma_b \cos \left( \frac{2\pi z}{d} \right) \right] P_2 (\cos \beta)$." [61].

Taking into account that from an order-of-magnitude point of view, there is a good agreement between theoretical predictions and experimental results for RSD coefficient in the bulk of the LC phase, this mean-field model lends credibility to the theoretical interpretation of the motional data and to the validity of that theoretical approach.

We conclude Section 2 by pointing out that the combination of the mean-field models with the experimental techniques provides a powerful tool for exploring and understanding the mechanisms which clarify the relevant underlying physical interactions.

## 3. Dynamics of the Layer-Thinning Processes in Free-Standing Smectic Films

The dynamic properties of LC systems confined within small spaces, such as the free-standing smectic films, are quite different from their bulk dynamics. A unique properties of FSSFs is their ability to demonstrate the layer-thinning transitions above $T_{\text{AI(N)}}$(bulk). Balance between surface and finite-sized effects leads to unusual layer-by-layer thinning when the interior layer(s) is(are) squeezed-out by the bounding ones. It has been assumed that the squeezing-out is initiated by a thermally activated nucleation process in which a density fluctuation forms a small hole of critical radius in the center of the circular smectic film [17]. If the hole inside the film is taken to be of circular shape with a radius $\epsilon$ and with a thickness of the order of the molecular length $d$, then under the effect of the pressure gradient $\nabla P$ one can develop the squeezing-out process between the squeezed-out and non-squeezed-out areas. In that case [12,17] $\nabla P$ is responsible for the driving out of one or several smectic layer(s) from the $N$-layer smectic film. The dynamics of the bounding area, which is separated by the layer-thinning transition front, during the layer-thinning transition $N \to N - 1$, will be described by using the conservation laws for mass and linear momentum [17,18], with and without accounting for the coupling between the smectic film and the meniscus.

### 3.1. Squeezing-Out Dynamics of Layer-by-Layer Thinning Transition in FSSF without Accounting for the Meniscus

"The evolution of the bounding area, which is separated by the layer-thinning transition front, from the $N$-layer to $(N - 1)$-layer smectic film, without accounting for the effect of meniscus, has been studied on the basis of conservation laws for mass and linear momentum [17]. The dynamics of the bounding area between the squeezed-out and non-squeezed-out areas in the FSSF has been investigated for the case of the circular shape with the area $A_0 = \pi R^2$, where $R$ is the radius of the total smectic film. In the framework of this approach the squeezing-out process starts from a small hole in the center of the circular smectic film. This hole is formed as a result of the thermally activated nucleation process in which a density fluctuation forms the small hole in the FSSF [17]. The evolution of the size of the small hole is determined by the balance between the bulk and surface thermodynamic forces, and the nucleus grows only when its size $\delta = \pi \epsilon^2 d$ exceeds a critical value $\delta_c$. Since the boundary between the squeezed-out and non-squeezed-out areas in the FSSF has been investigated for the case of the circular shape with the area $A(t) = \pi r^2(t)$, and the squeezing-out process continues until the area of the circle $A(t)$ reaches the value $A_0 = \pi R^2$, the shape of the bounding line can be treated as dislocation. Please note that during the thinning process all smectic layers are subjected to the compressive force $P$ acting across the FSSF upon heating to the isotropic temperature. This allows us to assume that the disjoining pressure [12,17,63] $P$ acting across the $N$-layer and $(N - 1)$-layer smectic film is responsible for develop of the pressure gradient $\nabla P$ between the squeezed-out and non-squeezed-out areas [12,17]. And since the disjoining pressure $P(N - 1)$ acting across the $(N - 1)$-layer film is greater then $P(N)$ acting across the $N$-layer smectic film [12,17,63], one can assume that the disjoining pressure (DP) is responsible for the pressure gradient $\nabla P$ which drives the squeezed-out smectic layer. All this allows us to assume that the conservation laws for mass and linear momentum must be held. Bearing in mind that the layer-thinning process in FSSF is characterized by removal of interior isotropic layer(s) from the overheated film, it should be taken into account only the continuity equation and the Navier–Stokes equation for the velocity field $\mathbf{v}(r,t)$. Taking into account the thickness of the smectic film, one can

assume, with high accuracy, that the mass density $\rho$ across the FSSF does not change, and one deals with an incompressible fluid. The incompressibility condition gives that [17]

$$\nabla \cdot \mathbf{v} = 0, \tag{34}$$

whereas the linear momentum balance equation can be written as [17]

$$\rho \frac{\partial \mathbf{v}(r,t)}{\partial t} = -\nabla_r P(r) + \nabla_r \sigma_{rz}(r,t), \tag{35}$$

where $\sigma_{rz}$ is the stress tensor component corresponding to the viscous force. In this case, it is convenient to choose a cylindrical coordinate system, where only one nonzero radial component $v(r,t)$ of the velocity vector $\mathbf{v}$, directed parallel to the smectic layers, exists

$$\mathbf{v} = v(r,t)\hat{\mathbf{e}}_r, \tag{36}$$

whereas the pressure

$$P = P(r), \tag{37}$$

is a function only of the radius $r$, and $\hat{\mathbf{e}}_r$ is the unit radial vector. The relevant solutions of Equations (34) and (35) can be written in the forms

$$v(r,t) = \frac{B(t)}{r}, \tag{38}$$

and

$$\sigma_{rz}(r,t) = \frac{1}{2}\alpha_4 \frac{\partial v(r,t)}{\partial r} = -\frac{1}{2}\alpha_4 \frac{B(t)}{r^2}, \tag{39}$$

respectively, and $\alpha_4$ is the shear viscosity coefficient. Substituting Equations (38) and (39) into Equation (35) gives [17]

$$\frac{1}{r}\frac{dB(t)}{dt} = -\frac{1}{\rho}\frac{\partial P(r)}{\partial r} + \frac{1}{2}\frac{\alpha_4}{\rho}\frac{B(t)}{r^3}. \tag{40}$$

This equation has a solution [17,64]

$$\ln\left(\frac{r}{R}\right)\frac{dB(t)}{dt} = -\frac{1}{\rho}\Delta P - \frac{\alpha_4}{4\rho}\frac{B(t)}{r^2}, \tag{41}$$

where the $\Delta P = P(N) - P(N-1)$ is a disjoining pressure dropping across the front of the moving boundary area during the layer-thinning transition $N \to N-1$. Taking into account that $A(t) = \pi r^2(t)$, the last equation can be rewritten as [17]

$$\frac{1}{2\pi}\ln\left(\frac{A(t)}{A_0}\right)\frac{d}{dt}\left[\frac{dA(t)}{dt}\right] = -\frac{2\Delta P}{\rho} - \frac{\alpha_4}{4\rho}\frac{1}{A}\frac{dA(t)}{dt}, \tag{42}$$

where

$$B(t) = \frac{1}{2\pi}\frac{dA(t)}{dt}.$$

In the following, the second time derivative term in Equation (42) has been neglected. Indeed, at the left end of the time's interval $[0, t_R]$, $r$ varies very slowly and the second-order time derivative of $r^2$ or $A(t)$ can be ignored, whereas at the right end of the same interval, $\lim_{t \to t_R} \ln\left(\frac{A(t)}{A_0}\right) = 0$. Here $t_R$ is the time which is needed to completely squeeze-out one

smectic layer. In these circumstances, the last Equation (42), which determines the value of the area $A(t)$ takes the form

$$\frac{dA(t)}{dt} + \frac{8\Delta P}{\alpha_4}A(t) = 0, \tag{43}$$

and its solution can be written as

$$A(t) = \kappa_c \exp\left[-\frac{8\Delta P}{\alpha_4}t\right], \tag{44}$$

where $\kappa_c = \pi\epsilon_c^2$, and $\epsilon_c$ is the radius of the critical nucleus. In turn, the expression for the velocity $v(r,t)$ can be written as

$$v(r,t) = -\frac{4\delta_c}{\pi}\frac{\Delta P}{\alpha_4}\frac{1}{r}\exp\left(-\frac{8\Delta P}{\alpha_4}t\right) = -\frac{4\epsilon_c^2\Delta P}{\alpha_4}\frac{1}{r}\exp\left(-\frac{8\Delta P}{\alpha_4}t\right), \tag{45}$$

where $r \in [\epsilon_c, R]$. The time $t_R$ which is needed to completely squeeze-out one smectic layer can be obtained from the relation $A_0 = \pi R^2 = \delta_c \exp\left(-\frac{8\Delta P}{\alpha_4}t_R\right)$, which gives

$$t_R = -\frac{\alpha_4}{4\Delta P}\ln\left(\frac{R}{\epsilon_c}\right). \tag{46}$$

Here we need to take into account the fact that the time $t_R$ is inversely proportional to $\Delta P = P(N) - P(N-1)$ and this value is always negative.

To examine the $\Delta P$'s effect on $t_R$ in the FSSF corresponding to the $N \rightarrow (N-1)$ layer-thinning transition, one need to have the values of the disjoining pressure $P(N)$ and $P(N-1)$, when the FSSF thins from $N$-layer to $(N-1)$-layer film. The numerical study of the disjoining pressure $P(N,T)$ of FSSFs for two cases, first, for the initially 25-layer film composed of the partially fluorinated 5-*n*-alkyl-2-(4-*n*-(*perfluoroalkyl-metheleneoxy*) *phenyl* (H10F5MOPP) molecules in air [12], and, second, for the initially 10-layer film composed of decylcyanobiphenyl (10CB) molecules in water [63], on heating to the isotropic temperatures has been carried out. Calculations showed that the layer-thinning transitions are characterized by abrupt (stepwise) rises to the higher values of $P(N-1)$ with respect to $P(N)$, when the film thins from $N$-layer to $(N-1)$-layer film, and all smectic layers, during the thinning, are subjected to the compressive force which grows with the number of $N$ as [12,63] $P(N) \sim \mathcal{O}\left(\frac{1}{N}\right)$. Such behavior of $P(N)$ provides the negative values of the DP [12,63] $\Delta P = P(N) - P(N-1)$. A number of data on the $\Delta P$ both in free-standing partially fluorinated H105FMOPP film in air [12], and in cyanobiphenyl 10CB film in water [63] is collected in Table 3 [17].

With Equation (46) and data on $\Delta P$ one can calculate the values of $t_R$ [in sec] and the average velocity $u = R/t_R$ [in m/sec]. These data also are collected in Table 3.

The calculation results collected in Table 3 correspond to the 25-layer FSSF composed of partially fluorinated H10F5MOPP molecules in air (The first seven lines from the top) [17], and to the 10-layer FSSF composed of 10CB molecules immersed in water (The fifth, forth, and third lines from the bottom), respectively. The measured data on the average thinning speed of single-layer thinning in 5-layer smectic film composed of H8F(4,2,1) MOPP molecules in air is given in two last lines [15].

**Table 3.** The calculated values of $\Delta P\,(N)$, $t_R$ and $u = R/t_R$ in FSSFs.

| | $-\Delta P\left[\frac{N}{m^2}\right]$ | $t_R\,[s]$ | $u\left[\frac{m}{s}\right]$ |
|---|---|---|---|
| Smectic 25 − layer film in the air [theor.] [10] | | | |
| $10 \rightarrow 9$ | $0.66 \times 10^3$ | $38.9 \times 10^{-5}$ | 0.26 |
| $9 \rightarrow 8$ | $2.64 \times 10^3$ | $9.7 \times 10^{-5}$ | 1.03 |
| $8 \rightarrow 7$ | $1.74 \times 10^3$ | $14.8 \times 10^{-5}$ | 0.67 |
| $7 \rightarrow 6$ | $3.98 \times 10^3$ | $6.5 \times 10^{-5}$ | 1.54 |
| Smectic 25 − layer film in the air [theor.] [13] | | | |
| $10 \rightarrow 9$ | $1.01 \times 10^4$ | $2.5 \times 10^{-5}$ | 3.93 |
| $9 \rightarrow 8$ | $0.5 \times 10^4$ | $5.1 \times 10^{-5}$ | 1.94 |
| $8 \rightarrow 7$ | $3.0 \times 10^4$ | $0.86 \times 10^{-5}$ | 11.63 |
| Smectic 10 − layer film in the water [theor.] [12] | | | |
| $9 \rightarrow 8$ | $1.65 \times 10^5$ | $1.35 \times 10^{-6}$ | 74.07 |
| | | | $u\,[\text{aver.}]\,\left[\frac{m}{s}\right]$ |
| Smectic 5 − layer film in air [exptl.] [24] | | | |
| $5 \rightarrow 4$ | | | 0.06 |
| $4 \rightarrow 3$ | | | 0.1 |

The results of disjoining pressure $\Delta P$ calculation performed using two different approaches [12] (see Table 3, for instance, the first three lines from (1) to (4)) (case I), and [19] (see Table 3, for instance, the lines from (5) to (7)) (case II)), (case II), gave results that differ from each other on average by one order of magnitude. Please note that the values of $\Delta P$ has been calculated for the same sequence of the layer-thinning transitions $10 \rightarrow 9 \rightarrow 8 \rightarrow 7$ in the FSSF composed of the same partially fluorinated H10F5MOPP molecules. As expected, the results of the time $t_R$ calculation performed using two different approaches (I) and (II) also gave results that differ from each other on average by one order of magnitude. The results of calculations also showed that the disjoining pressure $\Delta P$ values calculated for FSSF in water (case III) and air differ from each other by several orders of magnitude. Therefore, the values of $\Delta P$ which drives the squeezed-out one smectic layer in the case III are in two orders and one order higher than in the cases I and II, respectively [17]. This variation in pressure $\Delta P$ values leads to the fact that the time $t_R$ required for complete squeezing-out one smectic layer in case III is much less than in both cases I and II. The same situation is repeated when calculating the average velocities $u = R/t_R$, whose results are shown in the last column of Table 3 [17]. In all these calculations the value of $\alpha_4$ is equal to 0.1 [Pa s] and $R = 100$ [µm].

The condition for determining the value of the critical radius $\epsilon_c$ can be obtained by minimizing the energy $W$ required to form a small circular hole. The value of this energy $W$, formed by the three contributions, can be written in the form [17]

$$W\,(\epsilon) = 2\pi\,(N - n)\,d\gamma\epsilon - 2\pi\epsilon^2\gamma - \frac{1}{2}Bd\,(N - n)\,\pi\epsilon^2, \qquad (47)$$

where $\epsilon$ denotes the radius of the hole. Here, the first contribution is due to the line tension, while the second one is due to the interfacial contribution. Finally, the third term in Equation (47) is an elastic energy contribution. It should be noted that the first contribution to Equation (47) has a positive value, where $\gamma$ is the interfacial LC/air tension, $n$ is the number of squeezed-out layers, and $B$ is the compressional elastic constant which has the dimension of an energy per volume $V = \pi\epsilon^2\,(N - n)\,d$.

The last two terms give a negative contribution, where the first minus sign is due to the fact that in the process of creating of the hole, molecules are removed from the interface to the meniscus, whereas the second minus sign is due to the fact that in the process of creation of the hole the smectic film is compressed by the disjoining pressure. This contribution, due to the elastic force, will tend to

stabilize the hole. The condition for calculating the critical radius $\epsilon_c$ now takes the form $\partial W\left(\epsilon\right)/\partial\epsilon = 0$, which gives

$$\epsilon_c = \frac{2\left(N-n\right)d\gamma}{4\gamma + Bd\left(N-n\right)}. \tag{48}$$

When calculating the value of $\epsilon_c$ in the partially fluorinated H10F5MOPP 10-layer smectic film in air, we must have the following values: the values of the LC/air tension [11] $\gamma \approx 0.02$ [N/m], $d \approx 3$ [nm] and the values of $B = 10^7$ [N/m$^2$]. The results of calculations $\epsilon_c$, which corresponds to the layer-thinning transitions $10 \rightarrow 9 \rightarrow 8 \rightarrow 7 \rightarrow 6$, are as follows [17] (in [nm]): $\epsilon_c(10 \rightarrow 9) \approx 3.43$, $\epsilon_c(9 \rightarrow 8) \approx 3.38$, $\epsilon_c(8 \rightarrow 7) \approx 3.31$, and $\epsilon_c(7 \rightarrow 6) \approx 3.23$, respectively. For the decylcyanobiphenyl (10CB) molecules in water the value of the LC/water tension $\gamma$ is equal to 0.18 [N/m] [63], and $\epsilon_c(9 \rightarrow 8)$ is equal to 12.96 [nm] [17]. Calculations of dislocation loop kinetics in overheated FSSF in air, composed of H8F(4,2,1)MOPP molecules, showed that the velocity of thinning front is equal to $0.06 - 0.1$ [m/s] in 3-layer smectic film. That data on the velocity is also collected in the last column of Table 3 [17]. The evolution of the area of the circle $A\left(\tau\right)/A_0$ vs. $\tau = t/t_R(\text{max})$ [17], for the number of squeezing-out regimes in the FSSF in air: (1)-(10 $\rightarrow$ 9), (2)-(9 $\rightarrow$ 8), (3)-(8 $\rightarrow$ 7), and (4)-(7 $\rightarrow$ 6), respectively, is shown in Figure 27.

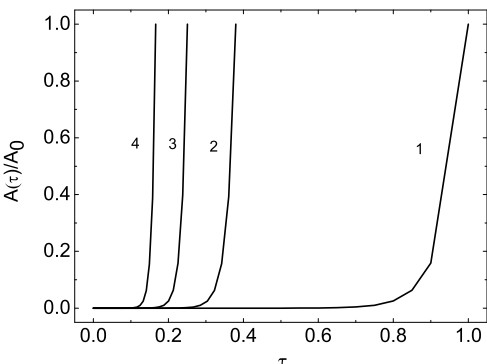

**Figure 27.** Plot of $A\left(\tau\right)/A_0$ vs. $\tau = t/t_R(\text{max})$ [17], for the number of squeezing-out regimes in the smectic film in air: (1)-(10 $\rightarrow$ 9), (2)-(9 $\rightarrow$ 8), (3)-(8 $\rightarrow$ 7), and (4)-(7 $\rightarrow$ 6), respectively.

Here $t_R(\text{max})$ is equal to the time $t_R$ corresponding to the $10 \rightarrow 9$ layer-thinning transition in FSSF composed of partially fluorinated H10F5MOPP molecules in air. The results of calculations of the circular area $A\left(\tau\right)/A_0$ show that the squeezing-out is accelerated at the final stage of the processes. Indeed, for instance, in the case of the layer-thinning transition $10 \rightarrow 9$ the value of the velocity $u\left(R,t_R\right)$, at the edge of the circular smectic film of the radius of $R$, is equal to 2.52 [m/s], what is on one order high than the value of the average velocity $u = R/t_R$, which is equal to 0.26 [m/s]. In all these calculations the value of the shear viscosity was equal to $\alpha_4 \approx 0.1$ [Pa s]. In turn, the results of experimental studies [65] indicate that the value of the rotational viscosity coefficient (RVC) $\gamma_1$ in the volume of the LC phase is very different from the value of RVC near the bounding surfaces. This fact must therefore be taken into account to obtain more reliable estimates of the shear viscosity coefficient $\alpha_4$.

It should also be noted that the value of the effective radius of nucleus has a strong influence on the dissipation at the smectic film/meniscus interface. Indeed, it has been shown that in smectic films the separated dislocations are coupled by means of the dissipation in the meniscus [66], and this dynamic coupling may change the effective radius of nucleus up to 10 times with respect to the static critical radius. As a result, the time required to completely squeeze-out one smectic layer in the smectic film will increase several times." [17].

### 3.2. Squeezing-Out Dynamics of Layer-by-Layer Thinning Transition in FSSF Accounting for the Meniscus

"In the previous paragraph we described the dynamics of squeezing-out of the number of smectic layers from the $N$-layer film without accounting for the effect of the meniscus. The evolution of the bounding area, which is separated by the layer-thinning transition front, from the $N$-layer to $(N-1)$-layer smectic film, without accounting for the effect of meniscus, has been studied on the basis of conservation laws for mass and linear momentum [17]. The dynamics of the bounding area between the squeezed-out and non-squeezed-out areas in the FSSF has been investigated for the case of the circular shape of the smectic film. This section will generalize the previous case to the case of accounting for the influence of the meniscus [17,18]. In this approach, the mechanism which is responsible for squeezing-out process, is based on the concept of the the disjoining pressure. In the previous paragraph, it was shown that the disjoining pressure $P(N)$ acting across the $N$-layer film is smaller than $P(N-1)$ acting across the $N-1$-layer smectic film [12,17]. As a result, it is formed the pressure gradient $\nabla P$ which drives the squeezed-out smectic layer in the zone far from the meniscus. We will assume that the influence of the meniscus extends only to the area $R - \delta \leq r \leq R$ closely adjacent to the interface between the FSSF and meniscus. As a result, the process of evolution of the bounding area will be affected by the additional pressure $P_1$ caused by the coupling of the smectic film with the meniscus. Here $\delta$ is the distance, counted from the smectic film/meniscus edge, where that effect occurs. All this allows us to assume that the conservation laws for mass and linear momentum must be held. Thus, the equation of the balance of linear moments acting on the unit volume of the smectic film, taking into account the influence of the meniscus, can be written as [18]

$$\ln y\,(\tau)\,\frac{d^2 y\,(\tau)}{d\tau^2} + \frac{2\pi^2}{y\,(\tau)}\left[\frac{\alpha_4 t_N}{2\rho A_0}\frac{dy\,(\tau)}{d\tau} + \left(\frac{dy\,(\tau)}{d\tau}\right)^2\right] + \frac{2\Delta\overline{P}}{\rho A_0}t_N^2 = 0, \tag{49}$$

where $\tau = t/t_N$ is the dimensionless time, $t_N$ is the normalization time, and $\Delta\overline{P}$ will be modeled by the linear function of the radius $r\,(t)$ as

$$\Delta\overline{P} = \begin{cases} \Delta P, & (\epsilon_c \leq r < R - \delta), \\ \frac{P_1}{\delta}\,(r\,(t) - R) + \Delta P + P_1, & (R - \delta \leq r \leq R). \end{cases}$$

Here $y\,(\tau) = A\,(\tau)/A_0$ [18] and $\Delta P = P(N) - P(N-1)$ is a disjoining pressure dropping across the front of the moving boundary area during the layer-thinning transition $N \to N-1$. If in the previous paragraph the value of disjoining pressure dropping across the moving front during the layer-thinning transition $N \to N-1$ was determined only by the disjoining pressure $\Delta P = P(N) - P(N-1)$, now an additional pressure $P_1$ acting from the meniscus on the smectic film should be accounted. Thus, the $P_1'$s effect will be extended to the submicrometer's distance $\delta$, and that effect will be modeled by the linear function $\Delta\overline{P}\,(r)$ of the distance $r$. Therefore, at $r = R$, $\Delta\overline{P}$ is equal to $P_1 + \Delta P$, whereas at $r = R - \delta$, $\Delta\overline{P}$ is equal to $\Delta P$. The justification of the choice of the linear form of the distance dependence of $\Delta\overline{P}$ is dictated by the submicrometer's range of $\delta$.

In the previous paragraph, without taking into account the influence of the meniscus, it was shown that Equation (42) can be simplified so that both the second time derivative term and the nonlinear term in Equation (42) [17] can be neglected. The neglecting of these two terms in Equation (49) was justified by the fact that the velocity **v** is small and at the left end of the time's interval $[0, t_R]$, $y\,(t)$ varies very slowly and one can ignore the second-order time derivative of $y\,(t)$, whereas at the right end of the same interval, $\lim_{t \to t_R} y\,(t) = 1$. Here $t_R$ is the time which is needed to completely squeeze-out one smectic layer. It should be noted that the time $t_N$ does not always coincides with the time $t_R$. Taking into account the above limitations, Equation (49), for the determination of $y\,(\tau)$, takes the form [18]

$$\frac{dy\,(\tau)}{d\tau} + 8y\,(\tau)\left[\lambda_1\left(\sqrt{y\,(\tau)} - 1\right) + \lambda_2\right] = 0, \tag{50}$$

where $\lambda_1 = \frac{t_R P_1 \epsilon_c}{\delta \alpha_4}$ and $\lambda_2 = \frac{t_R (\Delta P + P_1)}{\alpha_4}$ are two parameters of the smectic system, whereas $\epsilon_c$ is the radius of the critical nucleus. Linear Equation (50), with the initial condition $y(\tau = 0) = \left(\frac{\epsilon_c}{R}\right)^2$, has a solution

$$-4\tau = \frac{1}{\lambda_2 - \lambda_1} \ln \left[ \frac{\sqrt{y(\tau)}}{\lambda_1 \left( \sqrt{y(\tau)} - 1 \right) + \lambda_2} \left( \lambda_1 + \frac{R(\lambda_2 - \lambda_1)}{\epsilon_c} \right) \right]. \tag{51}$$

In this case, the relationship $y(\tau) = 1$ can be used for obtaining the value of the dimensionless time

$$-4\tau_R (P_1, \delta) = \frac{1}{\lambda_2 - \lambda_1} \ln \left[ \frac{1}{\lambda_2} \left( \lambda_1 + \frac{R(\lambda_2 - \lambda_1)}{\epsilon_c} \right) \right], \tag{52}$$

which is needed to completely squeeze-out one smectic layer accounting for the pressure $P_1$. In the limiting case $P_1 = 0$, when the effect of the meniscus on the smectic film is negligible, Equation (50) has the solution

$$y(\tau) = \frac{\exp\left[-8(\lambda_2 - \lambda_1)\tau\right](\lambda_2 - \lambda_1)^2}{\left[\lambda_1 \exp\left[-4(\lambda_2 - \lambda_1)\tau\right] - \mathcal{A}\right]^2}, \tag{53}$$

where $\mathcal{A} = \lambda_1 + \frac{R(\lambda_2 - \lambda_1)}{\epsilon_c}$ is the parameter of the smectic film.

It should be noted that when the influence of the meniscus can be ignored, the value of dimensional time $t_R (P_1 = 0) = -\frac{\alpha_4}{4\Delta P} \ln\left(\frac{R}{\epsilon_c}\right)$, calculated using Equation (52) coincides with the value of the time given in Equation (46)) [17]. In turn, the velocity $v(r,t)$ of the boundary between the squeezed-out and non-squeezed-out domains of the smectic film can be determined by using the equation

$$v(r,t) = \frac{1}{2\pi r} \frac{dA(t)}{dt}. \tag{54}$$

Thus, the velocity $v(r,t)$ is proportional to the first-order time derivative of $y(\tau)$ and inversely proportional to the radius $r(\tau)$.

Further detailed analysis will be given for evolution of the bounding area from the $N$-layer to $(N-1)$-layer smectic film during the layer-thinning process, based on accounting for both the second time derivative term and the nonlinear term in Equation (49) (case I), and when both these terms are neglected (case II) [18]. The results of calculations of the dimensionless squeezed-out area $y(\tau)$ (see Equation (49)) and the velocity $v(r,\tau)$ (see Equation (54)) of the bounding area between the squeezed-out and non-squeezed-out domains, accounting for the meniscus' effect will be presented. The effect of the disjoining pressure $\Delta P = P(N) - P(N-1)$ and the additional pressure $P_1$, caused by coupling of the smectic film with the meniscus, the distance $\delta$ and the radius of the critical nucleus $\epsilon_c$ on the nature of the squeezing-out dynamics, for the number of dynamic regimes, will be investigated. Calculations were performed for the following values of $\Delta P$ and $\epsilon_c$ [18]: $\Delta P = -0.66 \times 10^3$ N/m$^2$, $\epsilon_c = 3.43$ nm, for the case $10 \rightarrow 9$, $\Delta P = -2.64 \times 10^3$ N/m$^2$, $\epsilon_c = 3.38$ nm, for the case $9 \rightarrow 8$, $\Delta P = -3.98 \times 10^3$ N/m$^2$, and $\epsilon_c = 3.23$ nm, for the case $7 \rightarrow 6$, respectively. In all these calculations the value of $\alpha_4$ is equal to 0.1 Pa s and $R = 100$ μm.

In the case I, the nonlinear ordinary differential Equation (49) has been solved using the Runge–Kutta method of fourth order [67], and the result of calculation of the reduced area $y(\tau) = A(\tau)/A_0$ vs. $\tau = t/t_R(P_1 = 0)$, without accounting for the meniscus effect ($P_1 = 0$), are shown in Figure 28 (curves 1), for several layer-thinning transitions: $10 \rightarrow 9$ (Figure 28a), $9 \rightarrow 8$ (Figure 28b), and $7 \rightarrow 6$ (Figure 28c) [18], respectively. In the case II (curves 2), the reduced area $y(\tau) = A(\tau)/A_0$, calculated using Equation (53), as the function of the reduced time $\tau = t/t_R(P_1 = 0)$, are also shown in Figures 28. Here $\tau_R (I, II)$ is the time which is needed to completely squeeze-out one layer from the $N$-layer smectic film without accounting for the pressure $P_1$, i.e., the time when $y(\tau_R (I, II))$ is equal to 1. In this case $\tau_R (I)$ was obtained by means of numerical solution of Equation (49), whereas $\tau_R (II)$

was calculated using Equation (53) [18]. These calculations showed that the numerical result (case I) for evolution of $y(\tau)$ vs. $\tau$, for the layer-thinning transition $10 \to 9$, is faster approximately on the 20% with respect to the analytical result (case II) (see Figure 28a). It should be noted that at the final stage of the process of squeezing-out of the smectic layers, the results obtained by numerical methods and using analytical expressions are more and more close to each other results. For instance, in the cases $9 \to 8$ and $7 \to 6$ this difference almost disappears (see Figure 28b,c) [18].

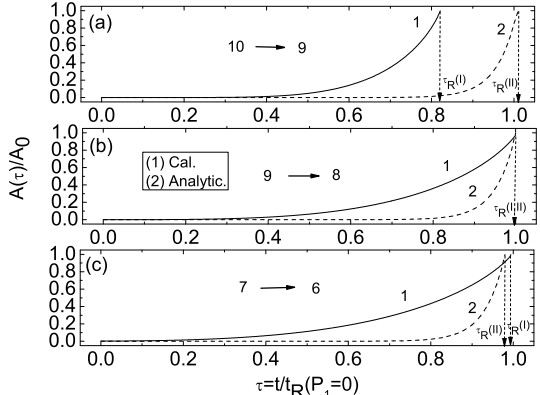

**Figure 28.** Plot of $A(\tau)/A_0$ vs. $\tau = t/t_R(P_1 = 0)$ [18] for two squeezing-out regimes in FSSF and for the case of $P_1 = 0$: (**a**)-($10 \to 9$), (**b**)-($9 \to 8$), and (**c**)-($7 \to 6$), respectively. Curves (1) (case I) are obtained by means of numerical solution of Equation (49), whereas curves (2) (case II) are calculated using Equation (53).

Having obtained the evolution of the $y(\tau)$ function, in the process of thinning the smectic film, one can calculate the velocity $v(\tau)$ vs. $\tau$, for two layer-thinning transitions: $10 \to 9$ and $7 \to 6$ regimes. The results of these calculations are shown in Figure 29 [18].

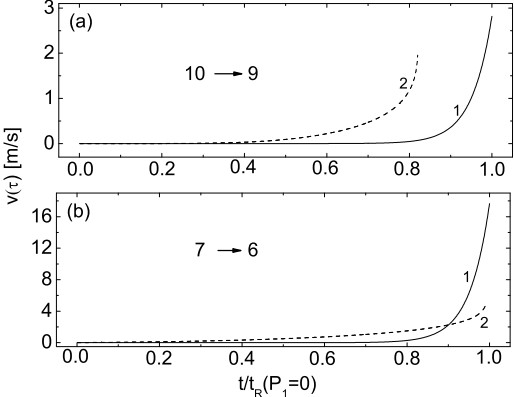

**Figure 29.** Plot of $v(\tau)$ vs. $\tau = t/t_R(P_1 = 0)$ [18], for the $10 \to 9$ (**a**) and $7 \to 6$ (**b**) squeezing-out regimes, respectively. Curves (1) and (2) correspond to cases I and II, respectively.

The curve (1) was calculated analytically using Equations (53) and (52), while the curve (2) was calculated numerically, using data on the function $y(\tau)$. Both results indicate that the dependence of the dimensionless velocity is characterized by the gradual increase in $v(\tau)$ with increasing of $\tau$. In addition, the results of the comparisons indicate that the influence of both the second time derivative term and the nonlinear term in the Navier–Stokes equation (Equation (49)) in the further calculations can be neglected.

The results of the calculation of the dynamics of thinning for the $10 \rightarrow 9$ squeezing-out regime, under the action of $\Delta \overline{P}$, accounting for the coupling of the smectic film with the meniscus, are shown in Figure 30a,b.

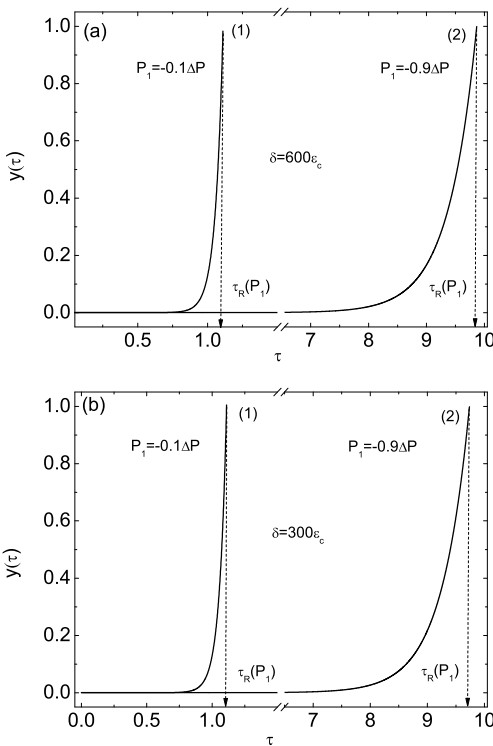

**Figure 30.** Plot of $y(\tau) = A(\tau) / A_0$ vs. $\tau = t / t_R (P_1 = 0)$ [18], for the $10 \rightarrow 9$ squeezing-out regime in the smectic film in air and for two values of $P_1$: (**a**) $-0.1 \Delta P$ (curve 1) and $-0.9 \Delta P$ (curve 2), respectively. Here $\delta = 600 \epsilon_c$. (**b**) Same as in (**a**), but for $\delta = 300 \epsilon_c$.

Figure 30a,b show the evolution of $y(\tau)$ vs. $\tau = t / t_R (P_1 = 0)$ for two values of the pressure $P_1$ acting on the smectic film, first, $P_1 = -0.1 \Delta P$ (curve 1) and, second, $P_1 = -0.9 \Delta P$ (curve 2), respectively. These calculations were performed for two values of distance $\delta$, first, for $\delta = 600 \epsilon_c (\sim 2 \, \mu m)$, second, for $\delta = 300 \epsilon_c (\sim 2 \, \mu m)$. The calculation results also showed that the meniscus has a strong effect on the time $t_R (P_1)$, whereas the distance $\delta$ practically does not affect that time. Indeed, for $P_1 = -0.9 \Delta P (\sim 0.6 \times 10^3 \, N/m^2)$ the value of the dimensionless time $\tau_R (P_1 = -0.9 \Delta P) (\tau_R (P_1 \neq 0) = t_R (P_1 \neq 0) / t_R (P_1 = 0))$ is equal to 9.88 $(\sim 4 \times 10^{-3} \, s)$ (Figure 30a (curve 2)), whereas for $P_1 = -0.1 \Delta P$, the value of the dimensionless time $\tau_R (P_1 = -0.1 \Delta P)$ is equal to 1.11 (Figure 30a (curve 1)), which is practically 9 times less. In all these cases the distance $\delta$ is equal to $600 \epsilon_c (\sim 2 \mu m)$. In the case when the distance $\delta$ decreases in 2 times, from $\delta = 600 \epsilon_c$ to $300 \epsilon_c$, the value of the time $\tau_R (P_1 = -0.9 \Delta P, \delta = 600 \epsilon_c) \approx 9.88$ (Figure 30a (curve 2)), which is practically the same as in the case of $\tau_R (P_1 = -0.9 \Delta P, \delta = 300 \epsilon_c) \approx 9.76$ (Figure 30b (curve 2)). The same tendency is kept for the case when the pressure $P_1$ is reduced by 9 times, from $P_1 = -0.9 \Delta P$ to $-0.1 \Delta P$. In this case, the time $\tau_R (P_1 = -0.1 \Delta P, \delta = 600 \epsilon_c) \approx 1.11$ (Figure 30a (curve 1)) which is the same as in the case of $\tau_R (P_1 = -0.1 \Delta P, \delta = 300 \epsilon_c) \approx 1.11$ (Figure 30b (curve 1)). These calculations showed that the influence of the meniscus is significant only when $P_1 \rightarrow -\Delta P$, and can be taken into account only at distances $\delta$ being of a few percent of the value of $R$. It is clearly seen that the evolution of the $A(\tau) / A_0$ profile is slowed down only at the final stage of the squeezing-out processes, when the values of the pressure $P_1$ reach the values of $-\Delta P$. Taking into account the fact that the value of $\Delta P = P(N) - P(N-1)$ is negative, one can conclude that the positive value of $P_1 = -\Delta P$ completely

slows down the squeezing-out process, whereas the negative values of $P_1$ lead to accelerating of the squeezing-out process during the layer-thinning transition.

Therefore, on the basis of this dynamic model of the squeezing-out process, everyone can make the conclusion that the external pressure, caused by coupling the smectic film with the meniscus, has a strong effect on that process and this dynamic coupling may significantly change the time which is needed to completely squeeze-out one or several layer(s) from the free-standing smectic film. Having obtained the data on $t_R (P_1 \neq 0)$, one can calculate the average velocity $u = R/t_R (P_1 \neq 0)$ [in m/s]. For instance, in the case of $10 \to 9$ transition and $P_1 = -0.9\Delta P$, the value of the time $t_R (P_1 = -0.9\Delta P)$ is equal to $9.88 \times t_R (P_1 = 0)$. Taking into account that $t_R (P_1 = 0)$, for the case of $10 \to 9$, is equal to $38.9 \times 10^{-5}$ [s] (see Table 1, Ref. [17]), one can estimate the average velocity $u (R, P_1 = -0.9\Delta P)$, at the edge of the circular smectic film of the radius $R = 100$ µm, as 0.026 m/s, what is in agreement with the value of the velocity of the thinning front $\sim 0.06$ m/s, obtained by means of the video measurements in overheated free-standing smectic film in air [15].

Thus, the results of calculations performed in the framework of the abovementioned dynamic model, which takes into account the influence of the meniscus on the smectic film, showed that the pressure gradient $\nabla P$ which develops between the squeezed-out and non-squeezed-out areas is responsible for the successive removal of one or several layer(s) from the $N$-layer smectic film during the layer-thinning process. It has been assumed that the squeezing-out is initiated by a thermally activated nucleation process in which the density fluctuation forms a small circular hole (void) of critical radius in the center of the circular smectic film. The origin of $\nabla P$ is a disjoining pressure (DP) acting across the $N$-layer and $(N - 1)$-layer smectic film, respectively. Taking into account the additional pressure $P_1$, which is responsible for coupling of the smectic film with the meniscus, a more realistic description of the thinning process in free-standing smectic film, when the temperature is slowly increased above $\theta_{\mathrm{AI}}$(bulk), has been proposed. In the framework of this model it was shown that the time $t_R (P_1 \neq 0)$ which is needed to completely squeeze-out one layer from the $N$-layer smectic film is inversely proportional to $\Delta P + P_1$. Bearing in mind that the value of $\Delta P = P(N) - P(N - 1)$ is negative, one can conclude that the positive value of $P_1 = -\Delta P$ completely slows down the squeezing-out process, whereas the negative values of $P_1$ lead to accelerating of the squeezing-out process during the layer-thinning transition." [18].

It should be noted that there is not yet a clear consensus on the mechanisms by which the layer-thinning occurs. Different mean-field theories have been used to obtain a qualitative description of the layer-thinning transitions [7,8,11,12,14,16,19,63], but the theoretical thinning temperatures were much larger than the experimental values [1]. Common features of all these theories are the existence of enhanced smectic ordering at the free surfaces of the film and the fact that thinning occurs when the smectic ordering in the interior of the film becomes sufficiently weak. Apart from details of the models used, the main differences among the theories are in the description of the kinetic processes by which layer-thinning occurs, i.e., whether this is by uniform squeezing-out of the melted interior [17,18], or via spontaneous nucleation of dislocation loops between domains of differing thickness [13,14,66]. Another mean-field theory [16], based on the generalization of the de Gennes model for a "presmectic" fluid confined between two solid walls, by means of including a quadratic term in the surface smectic OP while neglecting the external field term, also presents a simple analytical formula for variation of $T_{AI}(N)$ with $N$. Hence, further study on a wider range of compounds will be required to sort through the correlation between the transition temperatures resulting from the mean-field approaches and experimental measurements.

Please note that only the first set of the mean-field approaches [7,8,11,12,19–21] provides us an opportunity to calculate the disjoining pressure which is responsible for setting up of the pressure gradient, which drives the squeezed-out smectic film.

## 4. Conclusions

In this review, some recent progress made in the area of predicting structural and dynamic behavior associated with thin smectic films, both deposited on a solid surface or stretched over an opening when the temperature is slowly increased above the bulk transition temperature towards either the nematic or isotropic phases, has been discussed. The theoretical treatments for both of dynamic and static processes of flexible molecules in thin smectic films require a certain number of simplifying assumptions, which may only be justified by comparison between model predictions and experimental results. For instance, according to the set of mean-field theories followed here, thinning takes place when the smectic layer structure throughout the middle of the film vanishes. In an alternative theory, supported by experimental study, layer-thinning occurs in compounds which undergo first-order SmA-I transitions by spontaneous nucleation of dislocation loops, the growth of which causes a film to thin. A model of this thinning, predicting a layer-thinning transition temperature $T_{AI}(N)$ dependence, is functionally different from the power-law relation first described in [1] but fits experimental data closely. Another mean-field theory, based on the generalization of the de Gennes model for a "presmectic" fluid confined between two solid walls by means of including a quadratic term in the surface smectic OP while neglecting the external field term, also presents a simple analytical formula for variation of $T_{AI}(N)$ with $N$ which also fits experimental data very closely. Hence, further study on a wider range of compounds will be required to sort through the correlation between the transition temperatures resulting from the mean-field approaches and experimental measurements.

Thus, the combination of experimental techniques, such as the optical and calorimetric measurements, and theoretical approaches, based on the extended McMillan's theory, provides a powerful tool for investigating both the structural and dynamic properties of real smectic films, deposited on a solid surface or stretched over an opening.

**Author Contributions:** I.Ŝ.: writing-original draft preparation and editing. A.V.Z.: writing-original draft preparation and editing; supervision. All authors have read and agreed to the published version of the manuscript.

**Funding:** This research received no external funding.

**Conflicts of Interest:** The authors declare no conflict of interest.

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
