# Peer review of "Structural, Optical and Dynamic Properties of Thin Smectic Films"

_crystals, doi:10.3390/cryst10040321_

Round 1

Reviewer 1 Report

First of all, dynamics of free-standing LC films is far from my research area centered on molecular self-assembly and material structure/morphology, in bulk and thin film state. Therefore, my report only gives few general comments, without going in details of reference choices and pertinent aspects to review.

Free-standing films are indeed an academic subject by its own; what is missing for general reader is a small paragraph introducing the early development on the theme in the 80's-90's, for instance the seminal works of Sirota and Pershan, and of Pieranski.

More importantly, on page 4 authors argue that studies on free-standing LC films is a subject connected to organic thin film devices as used in organic light emitted diodes or organic solar cells. In devices, the active layers are however solid thin films mostly deposited from solution or by condensation from vapor phase. It is not evident how dynamics on liquid LC films could effectively contribute understanding and controlling the device film formation. By developing ideas about this analogy, this manuscript might become of interest for a broader scientific community. Otherwise, this digression is senseless for reader and authors should focus on the free-standing films theme.

Reviewer 2 Report

The submitted review article is entitled ‘Structural, optical and dynamical properties of thin smectic films’. The free-standing smectic films (FSSF) and lipid membranes are one of the crucial research topics in present-day soft matter science. Generally, they can be considered as a new class of two-dimensional state embedded into three-dimensional space in the long scale limit. The authors focus on the fluid-like films composed of nano-sized sheets of liquid crystal.

In the introduction section, the Authors provide state of the art for the presented topic. The introduction includes the most important cases. Then, the Authors presents studies of static properties of free-standing smectic films and molecular ordering in thin liquid crystal films on a solid surface and dynamics of the layer-thinning processes in free-standing smectic films. For this purpose, statistical-mechanical theories and optical and calorimetric techniques were combined.

The Authors discuss several crucial aspects like layer − thinning transitions and optical reflectivity, formulation of the mean-field model for the LC system confined in a small volume, the surface effect on the molecular ordering in thin LC systems on a solid surface, finite-size effect in thin LC films on a solid surface, mean-field theory with anisotropic forces for the description of the layer − thinning transition in FSSFs, layer − thinning transitions in free-standing partially fluorinated smectic films, heat capacity, surface tension, disjoining pressure and optical reflectivity of FSSFs, translational and orientational diffusion across the smectic films, dynamics of the layer-thinning processes in FSSFs. In addition, the Authors show some perspectives for further studies like mean-field theory based on the generalization of the de Gennes model for a ”presmectic” fluid confined between two solid walls.

The review partially bases on the previous peer-reviewed Authors’ papers. It is very valuable that they collect, compare and discuss the most important achievements in the form of open access review article for a broad audience. The selection of the references is appropriate. In conclusion, the authors show useful routes not only for further examining the validity of the theoretical description of thin SmA films but also for analyzing their structural, optical, thermodynamical and dynamical properties.

My remark is that despite most of the presented compounds are well known it is desirable to present chemical formulas/molecular structures of investigates liquid crystalline substances. This will make the story more attractive to follow especially in the section when the Authors discuss details of the molecular structures.

Reviewer 3 Report

The article presents an in-depth theoretical review of the models on the properties of thin smectic films. All models are well documented and explained, making it an interesting job. There are many calculated data sets and some explanations of the results. The authors draw conclusions which require a comparison with the experimental results.   I would suggest that the authors provide in more detail or in another article the user-friendly models and codes they used to attract the experimenters.
